# Integration-free Kernels for Equivariant Gaussian Process Modelling

**Tim Steinert** [1]  **David Ginsbourger** [1]  **August Lykke-Møller** [2]  **Ove Christiansen** [2]  **Henry Moss** [3][4]

## Abstract

We study the incorporation of equivariances into vector-valued GPs and more general classes of random field models. While kernels guaranteeing equivariances have been investigated previously, their evaluation is often computationally prohibitive due to required integrations over the involved groups. In this work, we provide a kernel characterization of stochastic equivariance for centred second-order vector-valued random fields and we construct integration-free equivariant kernels based on the notion of fundamental regions of group actions. We establish data-efficient and computationally lightweight GP models for velocity fields and molecular electric dipole moments and demonstrate that proposed integration-free kernels may also be leveraged to extract equivariant components from data.

## 1. Introduction

The incorporation of structural knowledge such as physical laws into machine learning models has gained significant attention for improving predictive accuracy and realism. For example, equivariances are ubiquitous across molecular chemistry, where applying simultaneous rigid motions on underlying atoms typically results in equivalent motions to their vectorial properties (as demonstrated in Figure 1). The incorporation of such equivariances is well-established in deep learning Cohen & Welling (2016), allowing neural networks to exploit knowing the responses across entire orbits of a group action from a single data point.

In contrast, progress incorporating such knowledge into Gaussian process (GP) models has been more moderate, potentially due to the intricate mathematical constraint of

[1]IMSV, University of Bern, Switzerland [2]Department of Chemistry, University of Aarhus, Denmark [3]School of Mathematical Sciences, Lancaster University, UK [4]Department of Applied Maths and Theoretical Physics, University of Cambridge, UK. Correspondence to: <tim.steinert@unibe.ch>.

*Proceedings of the 42nd International Conference on Machine Learning*, Vancouver, Canada. PMLR 267, 2025. Copyright 2025 by the author(s).

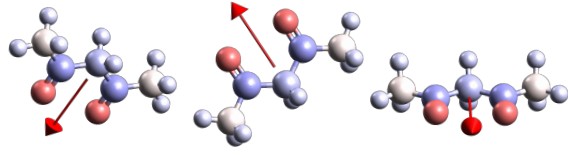

*Figure 1.* Rotational equivariance of electric dipole moments (red arrow) of acetylacetone molecules.

ensuring positive definiteness in matrix-valued covariance kernels. However, attracted by GP's explicit posterior distributions that facilitate uncertainty quantification and active learning, especially in scientific applications Moss et al. (2020); Griffiths et al. (2022); Ranković et al. (2024), there has been a significant recent effort to encode invariances and equivariances through tailored kernels Ginsbourger et al. (2012); Scheuerer & Schlather (2012); Ginsbourger et al. (2016); van der Wilk et al. (2018); Holderrieth et al. (2021); Henderson (2023).

Unfortunately, building equivariant kernels comes with significant computational effort, and choices are typically made that alleviate costs at the price of reducing expressiveness. Such computational challenges have been addressed in specific contexts, e.g., for accurate modelling of interatomic force fields Glielmo et al. (2017), by transforming a *scalar* argument-wise invariant kernel into an equivariant matrix-valued kernel with a single group integration.

Our work proposes a novel class of equivariant kernels that simultaneously overcome the previously high computational cost and limited flexibility of existing equivariant kernels. We exploit the group-theoretic notion of projecting onto fundamental regions of group actions, substantially extending the approach of Ginsbourger et al. (2012) as proposed for scalar-valued invariance. Our main contributions are:

1. a theoretical framework for stochastically equivariant random fields,

2. a class of integration-free equivariant kernels that are computationally efficient, flexible, and stable,

3. empirical results on equivariant fluid flows and dipole predictions from quantum chemistry.

## 2. Background

**Preliminaries** For a detailed summary of the necessary background on multivariate (Gaussian) random fields and group theory, we refer the reader to Appendix A. In what follows, we interchangeably denote vector-valued Gaussian random fields as GPs.

**Equivariances** In this work, we focus on efficiently encoding equivariances in random fields models. We consider $\mathbb{R}^p$-valued, second-order, centered random fields $Z = (\boldsymbol{Z_x})_{\boldsymbol{x} \in D}$, where $D \subset \mathbb{R}^d$. We denote by $\star$ a group action of a linear group $G$ on $D$. Equivariance of a mapping $f : D \to \mathbb{R}^p$ means that for any group element $g \in G$ and any $\mathbf{x} \in D$, the (vector) value taken by $f$ at the point $g \star \boldsymbol{x}$ is related to its value at $\boldsymbol{x}$ via multiplication with a matrix $\rho_g \in \mathbb{R}^{p \times p}$ representing $g$ in $\mathbb{R}^p$, i.e. $f(g \star \boldsymbol{x}) = \rho_g f(\boldsymbol{x})$.

In the context of a random field $Z$ we introduce a notion of *stochastic equivariance*, expressed as:

$$\forall g \in G, \; \boldsymbol{x} \in D, \quad \mathbb{P}(\boldsymbol{Z_{g\star x}} = \rho_g \boldsymbol{Z_x}) = 1. \quad (1)$$

Let us point out that for $G$ countable, $\forall g \in G$ can be brought inside of $\mathbb{P}$ without further assumptions, delivering a property of equivariance *up to a modification*, that is, $\forall \boldsymbol{x} \in D, \quad \mathbb{P}(\forall g \in G, \boldsymbol{Z_{g\star x}} = \rho_g \boldsymbol{Z_x}) = 1$. Adding conditions on $D$ (e.g., $D$ countable) similarly leads to a stronger notion of almost sure equivariance, namely that $\mathbb{P}(\forall g \in G, \boldsymbol{x} \in D, \boldsymbol{Z_{g\star x}} = \rho_g \boldsymbol{Z_x}) = 1$. Sufficient conditions for those types of equivariances in more general settings go beyond the scope of this work. In the following, we focus on stochastic equivariance.

Recall that a centred Gaussian random field $Z$ is characterized by its matrix-valued covariance kernel $K : D \times D \to \mathbb{R}^{p \times p}$, where for $\boldsymbol{x}, \boldsymbol{x}' \in D$ and $1 \le i \le p$,

$$K(\boldsymbol{x}, \boldsymbol{x}')_{ij} = \mathrm{Cov}\left(Z_{\boldsymbol{x}}^{(i)}, Z_{\boldsymbol{x}'}^{(j)}\right),$$

where the superscript $(i)$ refers to the $i$−th vector component. As we prove in Section 3, ensuring stochastic equivariance (1) for broad classes of random fields can be characterized in terms of a notion of kernel equivariance such as introduced in Reisert & Burkhardt (2007) for deterministic kernel-based algorithms.

**Related work** The Helmholtz kernel is currently considered a suitable kernel for GPs in the case $p = d = 2$, as it leverages the Helmholtz decomposition for vector fields defined over $\mathbb{R}^2$. While it ensures equivariance in the posterior mean (Prop. 4.2 Berlinghieri et al. (2023)), the Helmholtz kernel does not satisfy the requirements for stochastic equivariance. In the context of molecular properties, incorporating symmetries in GP models has been demonstrated to

be relevant Uteva et al. (2017). Symmetry-adapted GPs of Grisafi et al. (2018) effectively address equivariance but involve computationally expensive double sums. In the specific setting of O(3)-equivariance, Wigner kernels Bigi et al. (2024) offer an efficient approach to modelling covariances, built upon recursive constructions and analytical simplifications derived from properties of Wigner D-matrices. Conceptually aligned with our approach is the work of Aslan et al. (2023), which enforces equivariance in deterministic machine learning models within the setting of discrete groups by employing similar notions of fundamental regions and projections onto them.

## 3. Kernel characterizations of stochastic equivariance

We begin by characterizing in broad settings the stochastic equivariance of centred second order random fields in terms of their matrix-valued kernel. Theorem 3.1 lays the ground for the subsequent development of our computationally efficient kernel class.

**Theorem 3.1** (Kernel Characterization for stochastically equivariant random fields)**.** *Let $Z = (\boldsymbol{Z_x})_{\boldsymbol{x} \in D}, D \subset \mathbb{R}^d$, be a $\mathbb{R}^p$-valued square-integrable, centred random field with matrix-valued kernel $K : D \times D \to \mathbb{R}^{p \times p}$. Furthermore let $G$ be a linear group acting on $D$ via $\star$ and represented in $\mathbb{R}^p$ by $\rho : g \in G \to \rho_g \in \mathbb{R}^{p \times p}$. Then, the following equivalence holds:*

$$\forall \boldsymbol{x} \in D, \; g \in G, \;\; \mathbb{P}(\boldsymbol{Z_{g\star x}} = \rho_g \boldsymbol{Z_x}) = 1$$
$$\Longleftrightarrow$$
$$\forall \boldsymbol{x}, \boldsymbol{x}' \in D, \; g, h \in G,$$
$$K(g \star \boldsymbol{x}, h \star \boldsymbol{x}') = \rho_g K(\boldsymbol{x}, \boldsymbol{x}')\rho_h^{\top}. \quad (2)$$

**Proof.** Assuming $Z$ stochastically equivariant, we have that for any $\boldsymbol{x}, \boldsymbol{x}' \in D, \; g, h \in G$,

$$K(g \star \boldsymbol{x}, h \star \boldsymbol{x}') = \mathrm{Cov}(\boldsymbol{Z_{g\star x}}, \boldsymbol{Z_{h\star x'}})$$
$$= \mathbb{E}[\boldsymbol{Z_{g\star x}} \boldsymbol{Z_{h\star x'}}^{\top}] = \mathbb{E}[\rho_g \boldsymbol{Z_x}(\rho_h \boldsymbol{Z_{x'}})^{\top}]$$
$$= \rho_g \mathbb{E}[\boldsymbol{Z_x} \boldsymbol{Z_{x'}}^{\top}]\rho_h^{\top} = \rho_g K(\boldsymbol{x}, \boldsymbol{x}')\rho_h^{\top}.$$

Conversely, assuming (2), then, for any $\boldsymbol{x} \in D, \; g \in G$,

$$\mathrm{Cov}(\boldsymbol{Z_{g\star x}} - \rho_g \boldsymbol{Z_x}, \boldsymbol{Z_{g\star x}} - \rho_g \boldsymbol{Z_x})$$
$$= K(g \star \boldsymbol{x}, g \star \boldsymbol{x}) + \rho_g K(\boldsymbol{x}, \boldsymbol{x})\rho_g^{\top}$$
$$- K(g \star \boldsymbol{x}, \boldsymbol{x})\rho_g^{\top} - \rho_g K(\boldsymbol{x}, g \star \boldsymbol{x}) = \boldsymbol{0},$$

where from $\mathbb{E}(||\boldsymbol{Z_{g\star x}} - \rho_g \boldsymbol{Z_x}||^2) = \mathrm{tr}(\boldsymbol{0}) = 0$ and hence $\mathbb{P}(\boldsymbol{Z_{g\star x}} = \rho_g \boldsymbol{Z_x}) = 1$. $\square$

Property (2), i.e. for $K$ to be equivariant in the first argument and anti-equivariant in the second, is referred to in Reisert

& Burkhardt (2007) as (kernel) equivariance. In that sense, Theorem 3.1 establishes equivalence for centred second-order random fields between stochastic equivariance and equivariance of the underlying matrix-valued kernel. It is noticeable that Theorem 3.1 is quite general concerning the linear group $G$ and its action $\star$ on $D$. In particular, note that while we assume $D \subset \mathbb{R}^d$ throughout the paper, the result can be directly generalized to any $D$.

We now consider more specific assumptions. Following settings from Reisert & Burkhardt (2007), given a compact, linear and unimodular group $G$ with continuous representation in case of $G$ being a Lie group, such equivariant matrix-valued kernels can be constructed by Haar integration. Considering base matrix-valued kernels $K_o$ such that the integrals are well-defined, one obtains in fact as class of equivariant matrix-valued kernels by taking

$$K_\int(\boldsymbol{x}, \boldsymbol{x'}) = \int_{G^2} \rho_g^\top K_o(g \star \boldsymbol{x}, h \star \boldsymbol{x'}) \rho_h \, dg \, dh. \quad (3)$$

In particular, the equivariance of $K_\int$ can be checked by using the translation invariance of the Haar measure (defined up to a constant). Assuming further that the Haar measure is normalized, choosing any $K_o$ already satisfying equivariance in Eq. 3 leads to $K_\int = K_o$. In such settings, equivariant kernels can thus systematically be represented with this construction. In practice, however, the integral nature of Eq. (3) can make GP modelling with such kernels very computationally demanding.

## 4. Integration-free equivariant kernels

The primary limitation of modelling equivariant random fields with $K_\int$ is the need of evaluating the cumbersome double integral in Eq. (3) which is rarely available in closed form and must often instead be approximated, e.g., by quadrature methods. In addition, these approximations need to be accurate to allow the inversion of the Gram matrix required when fitting the GP. For instance, in the rotation-equivariant GP in Section 5.1, around 1000 function evaluations are needed, making posterior simulations over a few hundred locations computationally challenging.

The computational burden inherent to the group integration formulation of (3) motivate us to introduce a new class of integration-free kernels. Taking inspiration from a previously developed approach for scalar-valued random fields with invariant paths, as proposed by Ginsbourger et al. (2012), we propose projecting inputs onto a fundamental region of the group action, which is subsequently incorporated into a base matrix-valued kernel. Equivariance of the resulting kernel is enforced by left and right multiplication of the base kernel with suitable matrices following from the considered group representations.

As detailed in Appendix A, we denote by fundamental re-

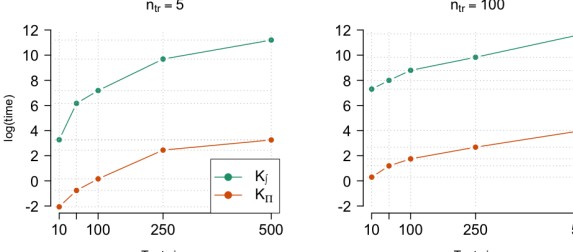

Figure 2. Log computation times (in seconds) for integration-based vs integration-free equivariant kernel evaluations.

gion of $\star$ a subset $A \subset D$ such that $G \star \bar{A} = D$, and $(g \star A) \cap A = \emptyset$, for any $g \in G \setminus \{e\}$. For such an $A$ and any $\boldsymbol{x} \in D$, there exists at least one $g \in G$ such that $g \star \boldsymbol{x} \in \bar{A}$. We call section any mapping $s \colon D \to G$, satisfying $s(\boldsymbol{x}) \star \boldsymbol{x} \in \bar{A}$, and denote by $\boldsymbol{\Pi}_s : D \to \bar{A}$ the associated projection onto $\bar{A}$, characterized by $\boldsymbol{\Pi}_s(\boldsymbol{x}) = s(\boldsymbol{x}) \star \boldsymbol{x}$ for $\boldsymbol{x} \in D$. The resulting class of kernels and their equivariance are presented in Proposition 4.1, followed by a worked example illustrating construction principles and resulting computational benefits.

**Proposition 4.1** (Integral-free equivariant kernels). *Let $G$ be a linear group acting on $D$ via $\star$, possessing a unitary group representation $\rho : g \in G \to \rho_g \in \mathbb{R}^{p \times p}$, and let $A \subset D$ be a fundamental region of $\star$. Then, for any matrix-valued kernel $K_{\bar{A}}$ on $\bar{A} \times \bar{A}$, section $s$ and associated projection $\boldsymbol{\Pi}_s$, $K_{\boldsymbol{\Pi}}$ below defines a matrix-valued kernel equivariant (w.r.t $\star$ and $\rho$) on $(G \star A) \times (G \star A)$:*

$$K_{\boldsymbol{\Pi}}(\boldsymbol{x}, \boldsymbol{x'}) = \rho_{s(\boldsymbol{x})}^\top K_{\bar{A}}(\boldsymbol{\Pi}_s(\boldsymbol{x}), \boldsymbol{\Pi}_s(\boldsymbol{x'})) \rho_{s(\boldsymbol{x'})}. \quad (4)$$

We refer to Appendix C for a proof of Proposition 4.1.

**Remark 4.2.** *In case of a free group action, $K_{\boldsymbol{\Pi}}$ is equivariant on the whole domain $D \times D$. Otherwise, $s(g \star \boldsymbol{x}) = s(\boldsymbol{x}) \circ g^{-1}$ may not hold for all $\boldsymbol{x} \in G \star \partial A$.*

**Example 4.3.** *Assume $D = \mathbb{R}^2$, $p = 2$, and $G = \mathrm{SO}(2)$ (An example of a $\mathrm{SO}(d)$-equivariant prediction task is presented in Appendix F). A fundamental region is then given by $A = \{(x, 0), x > 0\}$. To each point $\boldsymbol{x} \in D$, we assign a section $s(\boldsymbol{x})$ that rotates $\boldsymbol{x}$ into $\bar{A}$, represented for $\boldsymbol{x} \neq \boldsymbol{0}$ by:*

$$\rho_{s(\boldsymbol{x})} = \begin{bmatrix} \cos(\theta(\boldsymbol{x})) & -\sin(\theta(\boldsymbol{x})) \\ \sin(\theta(\boldsymbol{x})) & \cos(\theta(\boldsymbol{x})) \end{bmatrix}.$$

*Here, $\theta(\boldsymbol{x}) = -\arctan(x^{(2)}/x^{(1)})$ is the angle needed to rotate $\boldsymbol{x}$ onto $\bar{A}$ (clockwise). For $\boldsymbol{x} = \boldsymbol{0}$, we fix $\theta(\boldsymbol{0}) = 0$, since any rotation is a valid choice for $s(\boldsymbol{0})$.*

*The corresponding projection $\boldsymbol{\Pi}_s$ onto $\bar{A}$ maps each $\boldsymbol{x}$ to $\boldsymbol{\Pi}_s(\boldsymbol{x}) = (r(\boldsymbol{x}), 0)$, where $r(\boldsymbol{x}) = \|\boldsymbol{x}\|_2$. In Figure 2, we compare the total time to compute the posterior covariance matrices associated with $GP(0, K_{\boldsymbol{\Pi}})$ and $GP(0, K_\int)$ on*

*training and test locations in $[-1, 1]^2$, for different training and test set sizes. Here, $K_\int$ is computed by adaptive integration with the* `adaptIntegrate` *function in* R *on a maximum of* 1000 *function evaluations.*

*The base kernels $K_o = K_{\bar{A}}$ are chosen to be the simple diagonal RBF matrix-valued kernel. All computations in this paper were performed on a cluster equipped with single-core AMD EPYC2 CPUs running at a clock time of 2.25 GHz. With a straightforward implementation, the difference in computation time for moderate training and test set sizes is considerable. While computing the posterior distribution of $GP(0, K_\int)$ at 500 test locations given 100 training points requires 45 hours, with $GP(0, K_\pi)$ it takes a total of 55 seconds.*

**Proposition 4.4** (Continuity of $K_{\mathbf{\Pi}}$)**.** *Under the conditions of Proposition 4.1, assume that for a subset $B$ of $\bar{A}$, both $\rho_s$ and $\mathbf{\Pi}_s$ are continuous on $G \star B$, and $K_{\bar{A}}$ is continuous on $B \times B$. Then, $K_{\mathbf{\Pi}}$ is continuous on $(G \star B) \times (G \star B)$. In particular, for $B = \bar{A}$, $K_\Pi$ is continuous on $D \times D$.*

*Proof.* Assume $(\boldsymbol{x}, \boldsymbol{x}') \in (G \star B) \times (G \star B)$. Then, $(\mathbf{\Pi}_s(\boldsymbol{x}), \mathbf{\Pi}_s(\boldsymbol{x}')) \in B \times B$. Since $\mathbf{\Pi}_s$ is continuous on $(G \star B) \times (G \star B)$ and $K_{\bar{A}}$ is continuous on $B \times B$, it follows that $K_{\bar{A}}(\mathbf{\Pi}_s(\cdot), \mathbf{\Pi}_s(\cdot))$ is continuous on $(G \star B) \times (G \star B)$. By the continuity of matrix products of continuous matrix-valued functions, $K_{\mathbf{\Pi}}$ is continuous on $(G \star B) \times (G \star B)$.

If $B = \bar{A}$, then by definition of $A$, we have $G \star B = D$, which implies that $K_{\mathbf{\Pi}}$ is continuous on $D \times D$. $\qquad\square$

**Remark 4.5.** *If the continuity properties are not fulfilled with $B = \bar{A}$ but with $B = A$, this results in continuity on $(G \star A) \times (G \star A)$. Appendix B examines the continuity of $K_{\mathbf{\Pi}}$ in specific applications.*

## 5. Experiments

We now present a series of experiments designed to highlight the advantages of incorporating equivariances in GP models, alongside the specific benefits of our integration-free equivariant kernel. First, we model equivariant velocity fields, comparing with the popular Helmholtz kernel, which exhibits equivariance only in the posterior mean. Next, we consider a challenging real-world test case involving the prediction of molecular dipole moments, demonstrating the enhanced uncertainty quantification and practical applicability of our proposed kernel. Finally, we explore the efficacy of our GP models in a parameter estimation problem - disentangling a real-world ocean velocity dataset from equivariant perturbations.

### 5.1. Rotation-equivariant vector fields

**Data generation**   To assess the predictive performance of a zero-mean rotation-equivariant GP with our proposed integration-free kernel, we build a dataset of $n$ noisy measurements $\mathcal{D}^n = \{(\boldsymbol{x_i}, \boldsymbol{y_i})\}_{i=1}^n$, with $\boldsymbol{y}_i$ given as realisations of

$$\boldsymbol{F}(\boldsymbol{x_i}) + \boldsymbol{\varepsilon}_i, \quad \boldsymbol{\varepsilon}_i \sim \mathcal{N}(0, \sigma_{\text{obs}}^2 I_2)$$

for two synthetic $SO(2)$-equivariant vector fields

$$\boldsymbol{F}^1(\boldsymbol{x}) = (-\boldsymbol{x}^{(2)}, \boldsymbol{x}^{(1)}), \quad \boldsymbol{x} \in [-1, 1]^2,$$
$$\boldsymbol{F}^2(\boldsymbol{x}) = \frac{\boldsymbol{x}}{0.5 + \|\boldsymbol{x}\|^4}, \quad \boldsymbol{x} \in [-2, 2]^2,$$

with $n = 8, 10$ observations and $\sigma_{\text{obs}} = 0.15, 0.1$ for $\boldsymbol{F}^1$ and $\boldsymbol{F}^2$, respectively. See Appendix A.3 for a detailed explanation of the training and evaluation procedure.

**Baselines**   We consider four kernels constructed from a diagonal squared-exponential base kernel matrix function

$$K_{\text{SE}}(\boldsymbol{x}, \boldsymbol{x}'; \boldsymbol{\theta}) = \begin{bmatrix} \sigma_1^2 e^{-\|\boldsymbol{x} - \boldsymbol{x}'\|^2 / 2\ell_1^2} & 0 \\ 0 & \sigma_2^2 e^{-\|\boldsymbol{x} - \boldsymbol{x}'\|^2 / 2\ell_2^2} \end{bmatrix},$$

with tunable kernel parameters $\boldsymbol{\theta} = (\ell_1, \sigma_1, \ell_2, \sigma_2, \sigma_{\text{obs}})$. We build two $SO(2)$-equivariant kernels following the setup of Example 4.3, i.e. setting $K_o = K_{\bar{A}} = K_{\text{SE}}$, to produce our integral-free $SO(2)$-equivariant kernel $K_{\mathbf{\Pi}}$ and the double-integral $SO(2)$-equivariant kernel $K_\int$.

We also consider two additional kernels proposed for ocean modelling by Berlinghieri et al. (2023): (1) using $K_{\text{SE}}$ directly and (2) the Helmholtz matrix-valued kernel $K_{\text{H}}$, as derived by modelling the components $\Phi$ and $\Psi$ of a Helmholtz decomposition of a vector field $\boldsymbol{F} = \text{grad}\Phi + \text{rot}\Psi$ as independent GPs (see Berlinghieri et al. (2023) for a detailed derivation).

**Mean predictions**   The posterior mean predictions and root mean squared error (RMSE) over the ground truth field for the four GPs are shown in Figure 3, where we see that all kernels except $K_{\text{SE}}$ provide equivariant posterior means. However, we stress that the computational costs of our $GP(0, K_{\mathbf{\Pi}})$ were 500 times faster than those of $GP(0, K_\int)$, where the double integral (3) is approximated with the `adaptIntegrate` function in R using 1000 evaluations.

**Probabilistic predictions and sampling**   Figure 4 presents single realizations (samples) from the posterior distributions of each model, alongside the logarithmic posterior density of predictions over the ground truth field (LogS). Table 1 summarises average RMSE and LogS

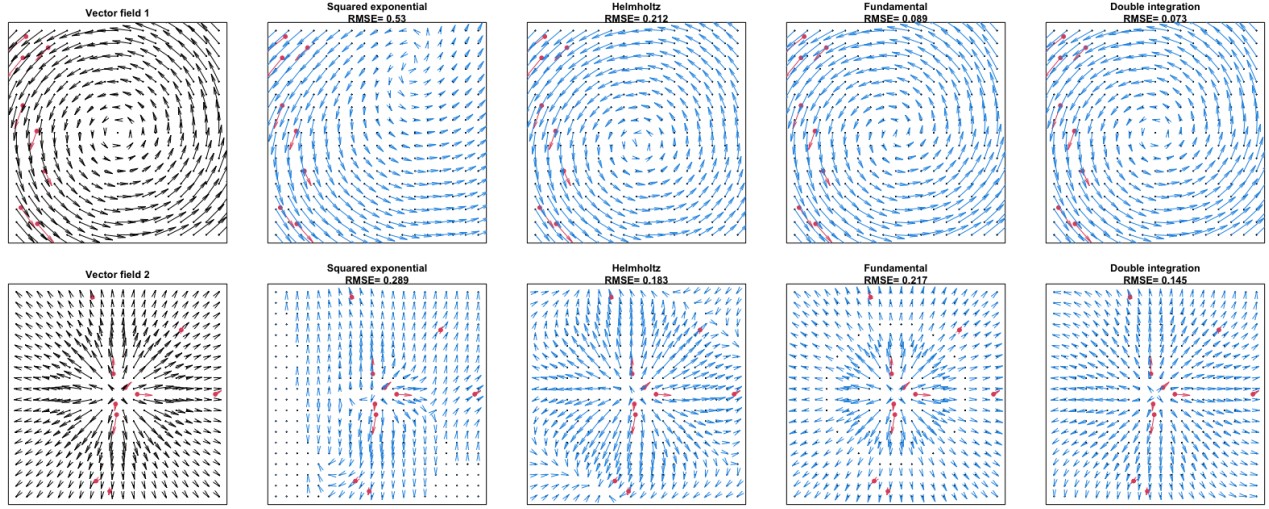

*Figure 3.* Ground truth (left), blue: posterior means of the squared exponential ($K_{SE}$), Helmholtz, fundamental ($K_{\Pi}$) and double integration ($K_{\int}$) GP. The top row corresponds to $\boldsymbol{F}^1$, the bottom row to $\boldsymbol{F}^2$ and red vectors indicate (noisy) observations.

scores for each 1000 random draws of $n = 8, 10$ training and $n_{\text{te}} = 17^2, 20^2$ test locations.

We see that the proposed integration-free kernel enjoys substantial advantages over all the other methods for two key reasons. Firstly, unlike the kernels $K_{\int}$ and $K_{\Pi}$ which satisfy our formal equivariance condition of Eq. (2), the Helmholtz GP does not provide rotation-equivariant posterior samples — see Corollary 5.1 below about posterior stochastic equivariance. Secondly, unlike our numerically stable $K_{\Pi}$, integral approximation errors when calculating $K_{\int}$ lead to instability along the the boundary of the domain.

**Corollary 5.1** (Posterior equivariance). *Assume a Gaussian random field $Z$ and a group $G$ satisfy the assumptions of Theorem 3.1, with the kernel of $Z$ being equivariant (2). Then, given any observed realization $z_{\text{tr}}$ of $\boldsymbol{Z}_{\text{tr}}$ (whereby the notation of Appendix A is used), the resulting posterior distribution retains stochastically equivariant, i.e.,*

$$\forall \boldsymbol{x} \in D, \ g \in G, \quad \mathbb{P}(\boldsymbol{Z}_{g \star \boldsymbol{x}} = \rho_g \boldsymbol{Z}_{\boldsymbol{x}} \mid \boldsymbol{Z}_{\text{tr}} = \boldsymbol{z}_{\text{tr}}) = 1.$$

**Remark 5.2.** *If a Gaussian random field $Z$ is stochastically equivariant on a subset of the domain $D$, then such posterior distributions retain stochastic equivariance on that subset. In particular, under the assumptions of Proposition 4.1, the resulting posterior distributions for a Gaussian process constructed via the kernel $K_{\Pi}$ retain stochastic equivariance on $G \star A$.*

**Remark 5.3.** *The cubic computational complexity of Gaussian process inference restricts its application to datasets of only a few thousand training and test points. Sparse approximations alleviate this limitation and enable GP modeling at larger scales. Interestingly, for usual constructions*

*Table 1.* Mean performance metrics [standard deviation]. Best scores are in bold, and values within the standard deviation of the best score indicated by (*).

| $\boldsymbol{F}$ | $K$ | $K_{\text{SE}}$ | $K_{\text{H}}$ | $K_{\Pi}$ | $K_{\int}$ |
|---|---|---|---|---|---|
| 1 | RMSE | 0.27 [0.11] | 0.23 [0.06] | 0.11 [0.06]* | **0.08** [0.04] |
| | LogS | 2.06 [137.88] | 88.44 [256.96] | **-6.03** [0.51] | -5.95 [0.46]* |
| 2 | RMSE | 0.33 [0.06] | 0.26 [0.06] | **0.13** [0.05] | 0.22 [0.2] |
| | LogS | 149.4 [1086.65] | 8.04 [84.35] | **-7.41** [0.81] | -3.41 [2.06] |

*of sparse GPs, building upon an equivariant kernel (and an equivariant mean) leads to sparse GP models retaining the equivariance properties in both their mean and covariance—thus preserving stochastic equivariance. We give further detail on this in Appendix G. This observation opens the door to extending equivariant GP modeling to large-scale datasets in the future.*

**Continuity of $K_{\Pi}$ and effect of $A$.** Downsides of $K_{\Pi}$ compared to $K_{\int}$ include potential discontinuities in $s$ and $\Pi_s$, potentially resulting in discontinuity of $K_{\Pi}$, as well as challenges with pathological choices of fundamental regions. To reduce boundary-related issues, one may favor connected fundamental regions. For example (See Fig. 5 for an illustration), partitioning $\{(x, 0), x > 0\}$ into $P = \cup_i P_i \times \{0\}$ with $P_i$'s intervals of equal length and defining $A = \cup_i (-1)^i P_i \times \{0\}$ as the fundamental region for the $SO(2)$-equivariant kernel in Example 4.3 leads to reduced performance. In Experiment 5.1, a partition of size 10 results in $GP(0, K_{\Pi})$ achieving an average RMSE (resp. LogS) score of $0.49$ (resp. $-2.56$) when predicting $\boldsymbol{F}_2$. Appendix E provides an illustration of this $A$ and furthermore the impact of a similarly ill-specified $A$ on the learning curves of Experiment 5.2.

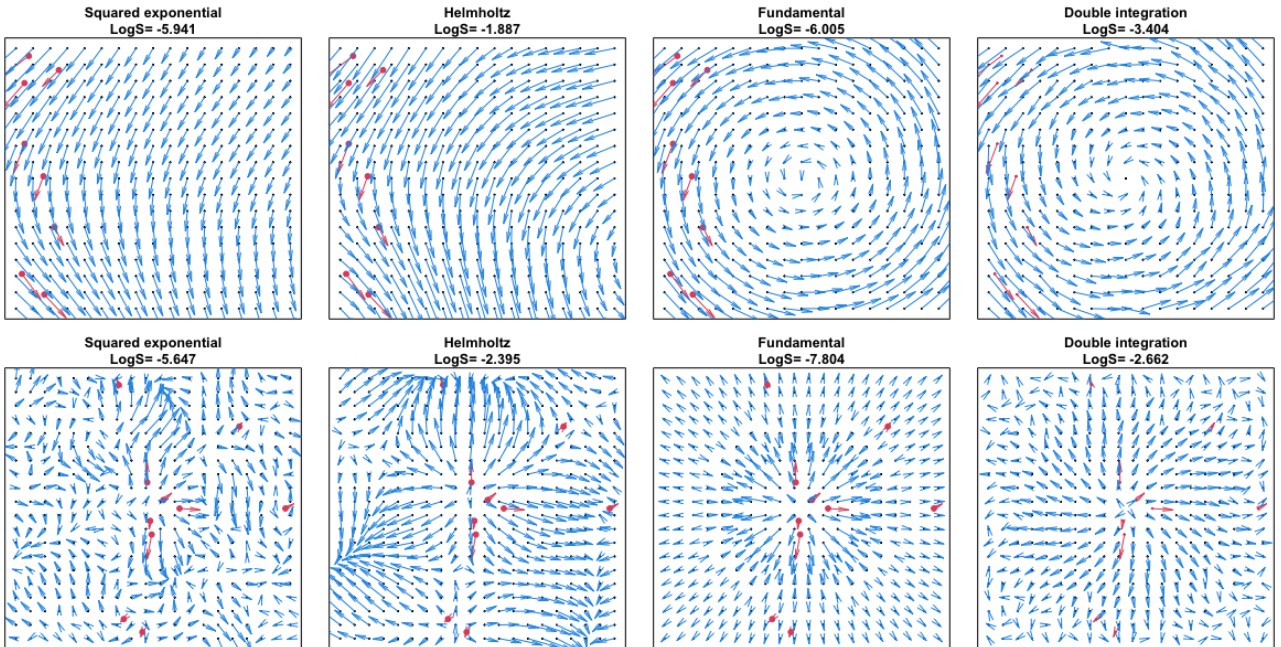

*Figure 4.* Single realizations of the Gaussian process posterior distributions. Unlike the Helmholtz and squared exponential GP posteriors, the two right-most models ensure equivariant realizations, as shown by Corollary 5.1

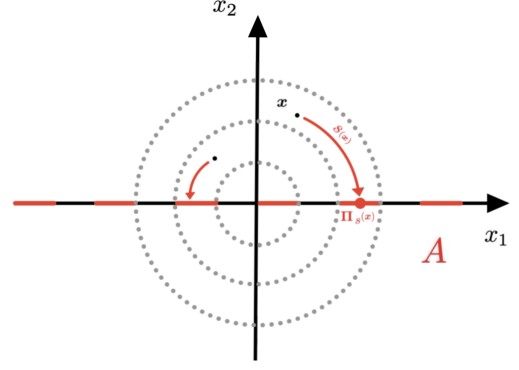

*Figure 5.* Visualisation of a disconnected fundamental region for Experiment 5.1 and the associated $s$ and $\mathbf{\Pi}_s$.

## 5.2. Water molecule dipole moments

The electric dipole moment, a vector indicating the imbalance in a molecule's electron distribution, is key to understanding intermolecular interactions Israelachvili (2011); Stone (2013) and predicting IR spectra intensities Califano (1976). Estimating the dipole moment across a molecular surface is computationally intensive, often requiring recalculations for numerous configurations, which can take days even for small molecules. Therefore, accurate statistical models are crucial, as they reduce the number of required calculations while still effectively describing the dipole surface, thereby making it feasible to predict IR spectra and

manage computational costs for larger molecules.

The electric dipole moment of a molecule is here modeled as a vector function $\boldsymbol{\mu} : \boldsymbol{x} \in D \subset \mathbb{R}^{3s} \to \boldsymbol{\mu}(\boldsymbol{x}) \in \mathbb{R}^3$, where $\boldsymbol{x} = \mathrm{Vec}\,(\boldsymbol{a_1}, \dots, \boldsymbol{a_s})$ encodes the position vectors in Euclidean space of the $s$ atoms ($\boldsymbol{a_i}, i = 1, ..., s$) within the considered molecule. From physical principles, $\boldsymbol{\mu}$ is known to be translation-invariant and rotation-equivariant, i.e. for all $\boldsymbol{x} \in D$,

$$\begin{cases} \boldsymbol{\mu}(\boldsymbol{t} \star_1 \boldsymbol{x}) = \boldsymbol{\mu}(\boldsymbol{x}) & \forall \boldsymbol{t} \in \mathbb{R}^3, \\ \boldsymbol{\mu}(g \star_2 \boldsymbol{x}) = \rho_g \boldsymbol{\mu}(\boldsymbol{x}) & \forall g \in \mathrm{SO}(3), \end{cases} \quad (5)$$

where $\star_1$ denotes the action of translations (encoded by elements of $\mathbb{R}^3$) on $\mathbb{R}^3$, $\star_2$ denotes the usual action of $\mathrm{SO}(3)$ on $\mathbb{R}^3$, and $\rho_g \in \mathbb{R}^{3\times3}$ is the rotation matrix (representation) canonically associated with $g \in \mathrm{SO}(3)$. The actions are extended to $D$ (and later to $\tilde{D}$) with the conventions:

$$\begin{cases} \boldsymbol{t} \star_1 \boldsymbol{x} = \mathrm{Vec}\,(\boldsymbol{t} \star_1 \boldsymbol{a_1}, \dots, \boldsymbol{t} \star_1 \boldsymbol{a_s}), \\ g \star_2 \boldsymbol{x} = \mathrm{Vec}\,(g \star_2 \boldsymbol{a_1}, \dots, g \star_2 \boldsymbol{a_s}). \end{cases} \quad (6)$$

We now provide an explicit example of how to apply our Eq. (4) to a quantum chemistry problem. In particular, we consider the case of water molecules, where $\boldsymbol{x} \in D \subset \mathbb{R}^9$ now represents the position vectors of an oxygen and two hydrogen atoms. In the considered case of water molecules where $s = 3$ and $\boldsymbol{a_2}, \boldsymbol{a_3}$ both stand for hydrogen atoms, there is also permutation-invariance with respect to these

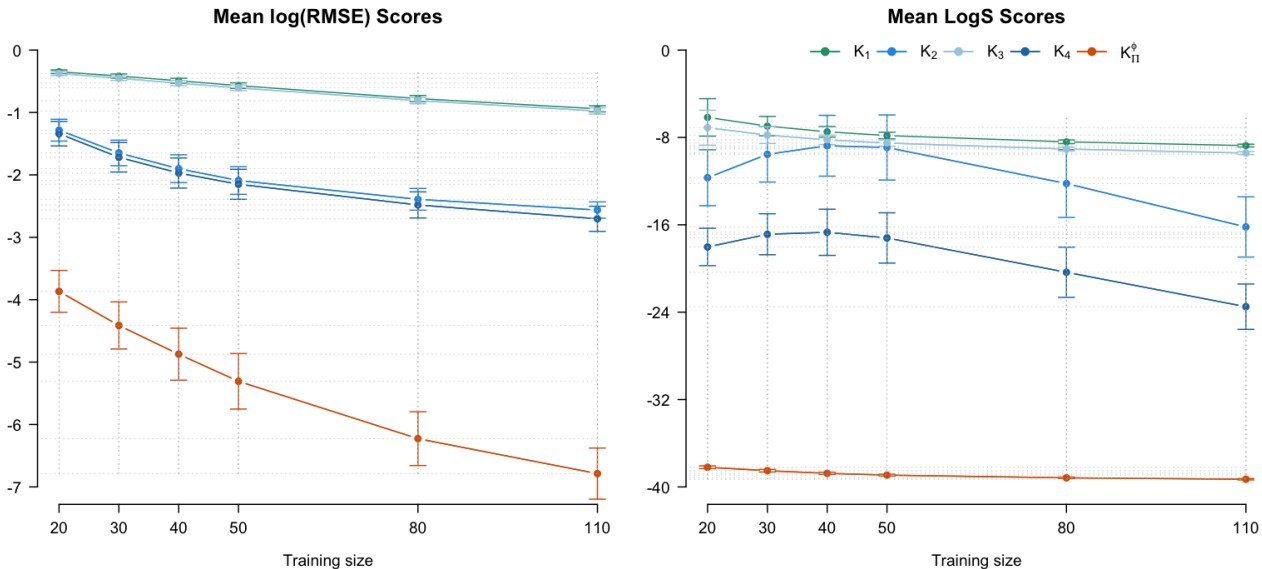

Figure 6. Predictive scores (mean $\pm$ sd over $10^3$ reps) of GP models versus training set size.

two columns. Specifically, for any $\boldsymbol{x} = \text{Vec}\,(\boldsymbol{a_1}, \ldots, \boldsymbol{a_s})$,

$$\boldsymbol{\mu}(\eta \star_3 \boldsymbol{x}) = \boldsymbol{\mu}(\boldsymbol{x}) \quad \eta \in \mathbb{Z}/2\mathbb{Z}, \tag{7}$$

where $\star_3$ stands for the action of $\mathbb{Z}/2\mathbb{Z}$ that swaps (for $\eta = \bar{1}$) the last and penultimate atom position vectors $\boldsymbol{a_s}$ and $\boldsymbol{a_{s-1}}$.

**Naive baseline** As a baseline kernel, we take a diagonal squared exponential matrix-valued kernel:

$$K_1(\cdot\,; \boldsymbol{\theta}) \colon D \times D \to \mathbb{R}^{3\times 3},$$

$$(\boldsymbol{x}, \boldsymbol{x'}) \mapsto \sigma^2 e^{-\frac{\|\boldsymbol{x} - \boldsymbol{x'}\|_2^2}{2\ell^2}} I_3,$$

where $\boldsymbol{\theta} = \left(\ell, \sigma^2\right)$ denotes the vector containing the tunable kernel lengthscale and variance hyperparameters. For ease of notation we write $K_1 = K_1(\cdot\,; \boldsymbol{\theta})$. We use a single lengthscale and variance, as there is no reason to assume different marginal variances for the components of $\boldsymbol{\mu}$, given that they all depend on the intrinsic geometry of the position vectors $\boldsymbol{x}$. Analogous experiments with separate lengthscales and variances did not lead to significant changes in the results.

**Constructing a tailored matrix-valued kernel** We assume that $\boldsymbol{\mu}(\boldsymbol{x}) = \boldsymbol{f}(\phi(\boldsymbol{x}))$, where $\boldsymbol{f} \colon \tilde{D} \to \mathbb{R}^3$ is a stochastically equivariant GP on $\tilde{D} \subset \mathbb{R}^6$. The map $\phi \colon D \to \tilde{D}$ is defined by $\phi(\boldsymbol{x}) = \eta(\Delta(\boldsymbol{x})) \star_3 \Delta(\boldsymbol{x})$ with $\boldsymbol{\Delta} \colon D \to \tilde{D}$ defined by $\boldsymbol{\Delta}(\boldsymbol{x}) = \text{Vec}\,(\bar{\boldsymbol{a}}_1, \bar{\boldsymbol{a}}_2) = \text{Vec}\,(\boldsymbol{a_2} - \boldsymbol{a_1}, \boldsymbol{a_3} - \boldsymbol{a_1})$ and $\eta \colon \tilde{D} \to \mathbb{Z}/2\mathbb{Z}$ is defined by $\eta(\bar{x}) = \bar{1}$ if $r_1 \geq r_2$ (and $\bar{0}$ otherwise), where $r_i = \|\bar{\boldsymbol{a}}_i\|$ ($i \in \{1, 2\}$). $\phi$ can be checked to be invariant under $\star_1$ and $\star_3$ and equivariant under $\star_2$. We model $\boldsymbol{f}$ as a centered GP

with equivariant kernel $K_{\boldsymbol{\Pi}}$, considering the fundamental region of $\star_2$ on $\tilde{D}$:

$$A = \left\{ \left( 0, \underbrace{\bar{a}_{12}}_{>0}, 0, \underbrace{\bar{a}_{21}}_{>0}, \underbrace{\bar{a}_{22}}_{\in \mathbb{R}}, 0 \right) : \bar{a}_{21}^2 + \bar{a}_{22}^2 < \bar{a}_{12}^2 \right\}.$$

To define a section for a point $\bar{\boldsymbol{x}} = \text{Vec}\,(\bar{\boldsymbol{a}}_1, \bar{\boldsymbol{a}}_2) \in \tilde{D}$, we apply a rotation $\Psi(\bar{\boldsymbol{x}}) \in SO(3)$ that maps $\bar{\boldsymbol{a}}_1$ to $(0, r, 0)$ with $r = \max\{r_1, r_2\}$, and $\bar{\boldsymbol{a}}_2$ to $(c_1, c_2, 0)$ with $c_1 > 0$. This rotation can be represented as a product of three elementary rotations in $SO(3)$: $\Psi(\bar{\boldsymbol{x}}) = \prod_{i=1}^3 \Psi_i(\bar{\boldsymbol{x}})$.

The corresponding projection map is given by

$$\boldsymbol{\Pi}_s(\bar{\boldsymbol{x}}) = (0, r, 0, c_1, c_2, 0).$$

We obtain $K_{\boldsymbol{\Pi}}$ on $\tilde{D}$ as

$$K_{\boldsymbol{\Pi}}(\bar{\boldsymbol{x}}, \bar{\boldsymbol{x}}') = \Psi(\bar{\boldsymbol{x}})^\top K_{\bar{A}}(\boldsymbol{\Pi}_s(\bar{\boldsymbol{x}}), \boldsymbol{\Pi}_s(\bar{\boldsymbol{x}}'))\Psi(\bar{\boldsymbol{x}}').$$

Finally, a kernel of $\boldsymbol{\mu}$ that is invariant under $\star_1$ and $\star_3$ and equivariant under $\star_2$, is given for $\boldsymbol{x}, \boldsymbol{x}' \in D$ by:

$$K_{\boldsymbol{\Pi}}^{\phi}(\boldsymbol{x}, \boldsymbol{x}') = K_{\boldsymbol{\Pi}}(\phi(\boldsymbol{x}), \phi(\boldsymbol{x}')).$$

An illustration of this procedure is shown in Figure 7. Note that the choices of $A$ and $\Psi$ are not unique. Here, $K_{\bar{A}}$ follows the same form as $K_1(\cdot\,; \boldsymbol{\theta})$, with inputs $\boldsymbol{\Pi}_s(\boldsymbol{\Delta}(\cdot))$ parameterized in (a subset of) $\mathbb{R}^3$ (by $r$, $c_1$, and $c_2$).

**Experimental Results** We consider a dataset of dipole moments $\boldsymbol{\mu}$ obtained from 850 water molecule configurations $\boldsymbol{x}$, which were computed using quantum chemical methods. For details on the dataset generation process, see

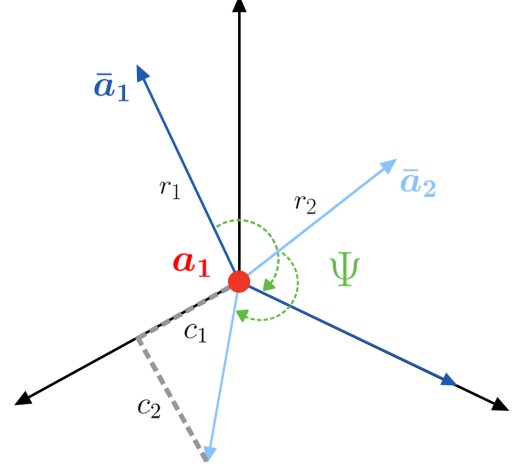

Figure 7. Illustration of the rotations for $K_{\boldsymbol{\Pi}}$.

Appendix H . To enhance the dataset's diversity and representativeness, we applied random rotations to the data points. See Figure 15 of the Appendix for a visualisation of this augmented dataset.

We compare the predictive accuracy of GP models for the dipole moments of water molecules using the baseline kernel $K_1$ and the proposed kernel $K_{\boldsymbol{\Pi}}$. See Appendix I for a visualisation of the optimised parameter values, sensitivity to initialisation, and additional information about our training schemes. For additional comparison, we can include invariances separately with the following kernels:

- **Translation-invariance:**

$$K_2(\boldsymbol{x}, \boldsymbol{x'}) = K_1(\boldsymbol{\Delta}(\boldsymbol{x}), \boldsymbol{\Delta}(\boldsymbol{x'})),$$

- **Permutation-invariance:**

$$K_3(\boldsymbol{x}, \boldsymbol{x'}) = K_1(\boldsymbol{\Pi}_{\star_3}(\boldsymbol{x}), \boldsymbol{\Pi}_{\star_3}(\boldsymbol{x'})),$$
$$\boldsymbol{\Pi}_{\star_3}(\boldsymbol{x}) = \eta(\boldsymbol{\Delta}(\boldsymbol{x})) \star_3 \boldsymbol{x},$$

- **Permutation and translation-invariance:**

$$K_4(\boldsymbol{x}, \boldsymbol{x'}) = K_1(\boldsymbol{\phi}(\boldsymbol{x}), \boldsymbol{\phi}(\boldsymbol{x'})).$$

In Figure 6, we see that incorporating structural knowledge into kernels consistently improves predictive accuracy. The proposed integration-free equivariant GP significantly outperforms the other GPs across all training set sizes, the small order of magnitudes of the RMSE suggest that our proposed model is accurate enough for use in quantum chemistry.

**Remark 5.4.** *The construction of the argument-wise rotation-equivariant and translation-invariant kernel $K_{\boldsymbol{\Pi}}$ for the dipole moment prediction task in ((5),(6)) transfers analogously to molecules of larger numbers of atoms $s$.*

*Preliminary learning curves in Appendix J on a newly obtained dipole moment dataset of 21,000 N-Methylformamide molecules of 9 atoms highlight the significantly improved predictive performance on test sets of size 500 of the equivariant Gaussian process over its base GP, particularly in data-scarce regimes ($n < 1000$) where structural priors are crucial. Sparse GP modeling on the full data set is part of ongoing work.*

**5.3. Ocean data with equivariant noise**

We finally investigate the performance of weighted combinations of (rotation-)equivariant and non-equivariant kernels on combinations of vector fields with equivariant perturbations in the case $d = p = 2$.

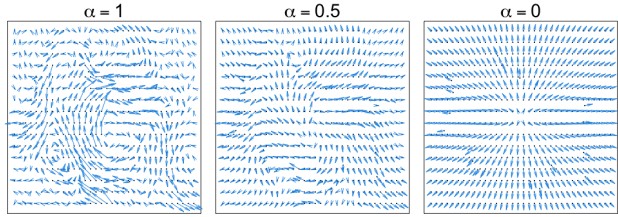

Figure 8. Gulf data with $SO(2)$-equivariant variation

In our experiment (see the visualization in Figure 8),

$$\boldsymbol{F}(\boldsymbol{x}) = \alpha \boldsymbol{F}^{\text{Ref}}(\boldsymbol{x}) + (1 - \alpha)\boldsymbol{F}^E(\boldsymbol{x}), \;\; \alpha \in [0, 1]. \quad (8)$$

$\boldsymbol{F}^{\text{Ref}}$ represents ocean drifter velocities on a set of 564 locations in the Gulf of Mexico, as taken from the Gulf Drifters Open dataset Lilly & Pérez-Brunius (2021), after standardizing $\boldsymbol{x}$ and $\boldsymbol{F}^G(\boldsymbol{x})$, and

$$\boldsymbol{F}^E(\boldsymbol{x}) = \frac{\boldsymbol{x}}{0.5 + \|\boldsymbol{x}\|^2}.$$

In Table 2 we compare model fits to $\boldsymbol{F}$, using 1000 replicates of 100-observation training sets and a 0.2 train/test ratio, of five different GPs:

1. $GP\left(0, K_{\text{SE}}\right)$,
2. $GP\left(0, \gamma^2 K_{\text{SE}} + (1 - \gamma)^2 K_{\boldsymbol{\Pi}}\right)$,
3. $GP\left(0, K_{\text{H}}\right)$,
4. $GP\left(0, \gamma^2 K_{\text{H}} + (1 - \gamma)^2 K_{\text{H}}^{\boldsymbol{\Pi}}\right)$, and
5. $GP\left(0, \gamma^2 K_{\text{H}} + (1 - \gamma)^2 K_{\boldsymbol{\Pi}}\right)$.

Here $K_{\text{H}}^{\boldsymbol{\Pi}}$ is the kernel matrix function which enforces the Helmholtz GP to be fully rotation-equivariant, obtained by taking $K_{\text{Helm}}$ as $K_o$ in the integration-free kernel of Example 4.3. The mixing coefficient $\gamma \in [0, 1]$ is an additional

Table 2. Mean performance metrics and [standard deviation]. Best scores are in bold, and values within the standard deviation of the best score given an asterisk (*).

| GP | $\alpha$ | 1 | 0.8 | 0.6 | 0.4 | 0.2 | 0 |
|---|---|---|---|---|---|---|---|
| 1 | RMSE | 0.787 [0.035]* | 0.630 [0.028]* | 0.476 [0.022]* | 0.323 [0.016] | 0.168 [0.010] | 0.021 [0.007] |
|   | LogS | 0.962 [0.082] | 0.035 [0.096] | -1.111 [0.122] | -2.617 [0.108] | -5.292 [0.085] | -18.872 [1.153] |
| 2 | RMSE | 0.766 [0.035]* | 0.617 [0.029]* | 0.468 [0.023]* | 0.317 [0.016]* | 0.166 [0.009] | 0.021 [0.014] |
|   | LogS | 0.857 [0.167]* | 0.003 [0.164]* | -1.121 [0.163] | -2.738 [0.162] | -5.420 [0.159] | **-20.317** [1.143] |
| 3 | RMSE | 0.787 [0.035]* | 0.629 [0.029]* | 0.471 [0.022]* | 0.316 [0.015]* | 0.162 [0.008]* | 0.015 [0.005]* |
|   | LogS | 0.932 [0.087]* | 0.001 [0.098]* | -1.201 [0.117] | -2.829 [0.133]* | -5.537 [0.140]* | -13.536 [0.022] |
| 4 | RMSE | 0.818 [0.050] | 0.644 [0.038] | 0.490 [0.030] | 0.340 [0.032] | 0.192 [0.026] | **0.011** [0.006] |
|   | LogS | 1.231 [0.343] | 0.264 [0.289] | -0.798 [0.265] | -2.258 [0.366] | -4.623 [0.545] | -13.763 [0.059] |
| 5 | RMSE | **0.761** [0.031] | **0.608** [0.023] | **0.458** [0.018] | **0.310** [0.012] | **0.158** [0.006] | 0.021 [0.004] |
|   | LogS | **0.773** [0.172] | **-0.137** [0.164] | **-1.302** [0.171] | **-2.898** [0.166] | **-5.541** [0.093] | -7.308 [0.005] |

kernel parameter optimised in the maximum likelihood setting, with initial value 0.5.

The results presented in Table 2 allow to compare models both in terms of point predictions (via the RMSE) and of probabilistic predictions (via the logarithmic score). Combinations of equivariant and non-equivariant kernels appear to consistently yield better performances than the use of single non-equivariant kernels alone. This stands out in particular for the logarithmic score, which underlines the importance of fine-tuning GPs in terms of distributional properties beyond the resulting posterior means. In this regard, the fifth combination, which stands out for most values of $\alpha$, appears as an intriguing blend between GPs equivariant in the mean and stochastically equivariant. Future numerical experiments will aim at exploring this and further combinations more extensively.

## 6. Conclusion and perspectives

We presented a theoretical framework for stochastically equivariant second order random fields and introduced a method to construct a class of equivariant random field models by leveraging fundamental regions, that avoids cumbersome group integration. It is worth noting that neither integration-based nor fundamental region approaches can be declared universally superior for stochastically equivariant GP modeling. While the former offers an elegant construction principle, the latter provides a fast and practical alternative that still satisfies the equivariance requirements and enables tackling challenging prediction tasks. Our experiments on rotation-equivariant synthetic and real-world data show that the proposed approach enables obtaining lightweight GP models honoring prescribed equivariances, leading to benefits both on the computational side and in terms of probabilistic prediction performance. Our approach was found in particular to allow for efficient predictions of dipole moments of water molecules by incorporating physical principles directly into the GP framework. It was also shown to allow for expressive kernel combinations and obtain competitive probabilistic predictions for ocean velocity data with equivariant perturbations.

Future work includes scaling up equivariant GP modeling to large molecule datasets like the full N-Methylformamide dataset using sparse equivariant GPs. Beyond scaling to larger data, we also aim to extend our approach to other natural and artificial systems, broadening the applicability of equivariant GP modeling in scientific research. Considering proposed kernels within wider machine learning pipelines providing prediction of molecular properties or active learning (e.g. Moss & Griffiths (2020); Griffiths et al. (2024)) could be of interest. Also, further exploring mathematical properties of those kernels and investigating potential synergies with recent developments pertaining to kernels on graphs, Lie groups, and other structures (See, e.g., Azangulov et al. (2024)) are of interest.

**Data availability and code** We provide the dipole moment data from Section 5.2 along with a notebook containing the code necessary to reproduce the experiments in this Github repository.

## Acknowledgments

The authors would like to thank all people having evaluated and provided feedback on this work for comments and suggestions having led to substantial improvements. Special thanks to Sebastian Baader and Jan Draisma for useful feedback pertaining to the terminology of fundamental regions. GP calculations were performed on UBELIX (https://www.id.unibe.ch/hpc), the HPC cluster at the University of Bern. Tim Steinert and David Ginsbourger acknowledge the support of the Digitization Commission (DigiK) of the University of Bern via the project "Perception in Statistics, Econometrics and Probability". Part of this research was performed while Tim Steinert and David Ginsbourger were visiting the Institute for Mathematical and Statistical Innovation (IMSI), which is supported by the

National Science Foundation (Grant No. DMS-1929348). David Ginsbourger would like to thank the Isaac Newton Institute for Mathematical Sciences, Cambridge, for support and hospitality during the program Representing, calibrating & leveraging prediction uncertainty from statistics to machine learning, where work on this paper was undertaken that was partially supported by EPSRC grant EP/Z000580/1 and by a grant from the Simons Foundation. The work of August Lykke-Møller and Ove Christiansen was supported by the Danish National Research Foundation through the Center of Excellence for Chemistry of Clouds (Grant Agreement No: DNRF172). Ove Christiansen also acknowledges support from the Independent Research Fund Denmark through Grant No. 1026-00122B. The research work of Henry Moss was supported through Schmidt Sciences, LLC and Lancaster University's Mathematics for AI in Real-world Systems E3 grant.

## Impact Statement

This paper presents work whose goal is to advance the field of Machine Learning. There are many potential societal consequences of our work, none which we feel must be specifically highlighted here.

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

# Appendix

## A. Preliminaries

### A.1. Multivariate random field

Let $(\Omega, \mathcal{F}, \mathbb{P})$ be a probability space. We denote by $\mathbb{R}^p$-valued random variable a measurable mapping

$$\boldsymbol{V} : (\Omega, \mathcal{F}) \to (\mathbb{R}^p, \mathcal{B}(\mathbb{R}^p)),$$

where $\mathcal{B}(\mathbb{R}^p)$ stands for the Borel sigma-algebra of $\mathbb{R}^p$. We call a $\mathbb{R}^p$-valued random variable $\boldsymbol{V}$ *square-integrable* whenever $\mathbb{E}(\|\boldsymbol{V}\|^2) < \infty$, where $\|\|$ stands for the Euclidean norm in $\mathbb{R}^p$. For such a square-integrable $\boldsymbol{V}$, there exist a vector $\boldsymbol{m} \in \mathbb{R}^p$ and a positive semi-definite matrix $K \in \mathbb{R}^{p \times p}$ such that

$$\mathbb{E}[\boldsymbol{V}] = \left(\mathbb{E}[V^{(i)}]\right)_{i=1}^p = \boldsymbol{m},$$

and

$$\mathrm{Cov}[\boldsymbol{V}] = \left(\mathrm{Cov}\left[V^{(i)}, V^{(j)}\right]\right)_{1 \le i, j \le p} = K,$$

where $V^{(i)}$ denotes the $i-$th component of the random vector $\boldsymbol{V}$.

Now, for some set $D$, a $\mathbb{R}^p-$valued random field is a collection of $\mathbb{R}^p-$valued random vectors defined on the probability space $(\Omega, \mathcal{F}, \mathbb{P})$, indexed over $D$, that we denote by $Z = (\boldsymbol{Z_x})_{\boldsymbol{x} \in D}$. We call $Z$ square-integrable, if for all $\boldsymbol{x} \in D$, $\boldsymbol{Z_x}$ is square-integrable in the above sense. Then there exist mappings $\boldsymbol{m} \colon D \to \mathbb{R}^p$ and $K \colon D \times D \colon \to \mathbb{R}^{p \times p}$ defining the expected value at any $\boldsymbol{x} \in D$ by

$$\boldsymbol{m}(\boldsymbol{x}) = \mathbb{E}[\boldsymbol{Z_x}]$$

and the matrix-valued kernel of $Z$ describing cross-covariances by

$$K(\boldsymbol{x}, \boldsymbol{x}') = \mathrm{Cov}(\boldsymbol{Z_x}, \boldsymbol{Z_{x'}}),$$

where $\boldsymbol{x}, \boldsymbol{x}' \in D$. The kernel $K$ needs to satisfy $K(\boldsymbol{x}', \boldsymbol{x}) = K(\boldsymbol{x}, \boldsymbol{x}')^\top$ (for any $\boldsymbol{x}, \boldsymbol{x}' \in D$) and be *positive-semi definite*, meaning that, for any $n \ge 1$, $\boldsymbol{a}_1, \dots, \boldsymbol{a}_n \in \mathbb{R}^p$, and $\boldsymbol{x_1}, \dots, \boldsymbol{x_n} \in D$, it holds

$$\sum_{1 \le i, j \le n} \boldsymbol{a}_i^\top K(\boldsymbol{x_i}, \boldsymbol{x_j}) \boldsymbol{a}_j \ge 0.$$

Throughout this work, we consider the index set $D$ to be a subset of $\mathbb{R}^d$, where $d \ge 1$.

### A.2. Multivariate Gaussian random field

$Z$ is said to be a multivariate Gaussian random field if for any $n \ge 1$, and $\boldsymbol{x_1}, \dots, \boldsymbol{x_n} \in D$, $\mathrm{Vec}\left(\boldsymbol{Z_{x_1}}, \dots, \boldsymbol{Z_{x_n}}\right)$ has a multivariate Gaussian distribution. In what follows, we think of $\boldsymbol{Z}_{\mathrm{tr}} = \mathrm{Vec}\left(\boldsymbol{Z_{x_1}}, \dots, \boldsymbol{Z_{x_n}}\right)$ as responses at set of training points denoted $X_{\mathrm{tr}} = [\boldsymbol{x_1}, \dots, \boldsymbol{x_n}] \in \mathbb{R}^{d \times n}$. Further denoting by $\boldsymbol{z}_{\mathrm{tr}}$ a vector standing for an observed realization of $\boldsymbol{Z}_{\mathrm{tr}}$, we use the notation

$$\mathcal{D}_n = \{(\boldsymbol{x_i}, \boldsymbol{z_{x_i}})\}_{i=1}^n.$$

While the $\boldsymbol{x_i}$ points are not considered as random here, we use a $\mathcal{D}_n$ subscript in forthcoming equations as a shorthand notation to summarize the information needed to condition $Z$ based on the considered $n$ training evaluations of $Z$ (i.e. to condition $Z$ on the event "$\mathrm{Vec}\left(\boldsymbol{Z_{x_1}}, \dots, \boldsymbol{Z_{x_n}}\right) = \boldsymbol{z}_{\mathrm{tr}}$").

Prior to any observation, the distribution of $\boldsymbol{Z}_{\mathrm{tr}}$ is $\mathcal{N}\left(\boldsymbol{m}\left(X_{\mathrm{tr}}\right), K\left(X_{\mathrm{tr}}\right)\right),$ where

$$\boldsymbol{m}\left(X_{\mathrm{tr}}\right) = \left(m^{(1)}(\boldsymbol{x_1}), \dots, m^{(1)}(\boldsymbol{x_n}), \dots, m^{(p)}(\boldsymbol{x_1}), \dots, m^{(p)}(\boldsymbol{x_n})\right) \tag{9}$$

and

$$K\left(X_{\mathrm{tr}}\right) = \begin{bmatrix} K_{11}\left(X_{\mathrm{tr}}\right) & \dots & K_{1p}\left(X_{\mathrm{tr}}\right) \\ \vdots & \ddots & \vdots \\ K_{p1}\left(X_{\mathrm{tr}}\right) & \dots & K_{pp}\left(X_{\mathrm{tr}}\right) \end{bmatrix} \in \mathbb{R}^{pn \times pn}. \tag{10}$$

For $1 \leq i, j \leq p$, the block matrices in (10) are given by

$$\mathbb{R}^{n \times n} \ni K_{ij}(X_{\mathrm{tr}}) = \left( (K(\boldsymbol{x}_l, \boldsymbol{x}_m))_{i,j} \right)_{1 \leq l, m \leq n}.$$

Similarly, the cross-covariance for any two sets of locations

$$X_1 = [\boldsymbol{x_1}, \ldots, \boldsymbol{x}_{n_1}] \in \mathbb{R}^{d \times n_1}, \quad X_2 = [\boldsymbol{x'}_1, \ldots, \boldsymbol{x'}_{n_2}] \in \mathbb{R}^{d \times n_2},$$

is given by

$$K(X_1, X_2) = \begin{bmatrix} K_{11}(X_1, X_2) & \ldots & K_{1p}(X_1, X_2) \\ \vdots & \ddots & \vdots \\ K_{p1}(X_1, X_2) & \ldots & K_{pp}(X_1, X_2) \end{bmatrix} \in \mathbb{R}^{pn_1 \times pn_2},$$

where

$$\mathbb{R}^{n_1 \times n_2} \ni K_{ij}(X_1, X_2) = \left( (K(\boldsymbol{x}_l, \boldsymbol{x'}_m))_{i,j} \right)_{1 \leq l \leq n_1, \, 1 \leq m \leq n_2}.$$

**Posterior distribution** For a centred Gaussian random field $Z$ and test locations $X_{\mathrm{te}}$, the posterior distribution of $\boldsymbol{Z}_{\mathrm{te}}$ (defined analogously to $\boldsymbol{Z}_{\mathrm{tr}}$ in terms of $X_{\mathrm{te}}$ instead of $X_{\mathrm{tr}}$) given $\mathcal{D}_n$ is thus characterised by the posterior mean

$$\boldsymbol{m}_{\mathcal{D}_n}(X_{\mathrm{te}}) = K(X_{\mathrm{te}}, X_{\mathrm{tr}}) K(X_{\mathrm{tr}})^{-1} \boldsymbol{z}_{\mathrm{tr}} \tag{11}$$

and the posterior covariance

$$K_{\mathcal{D}_n}(X_{\mathrm{te}}) = K(X_{\mathrm{te}}) - K(X_{\mathrm{te}}, X_{\mathrm{tr}}) K(X_{\mathrm{tr}})^{-1} K(X_{\mathrm{tr}}, X_{\mathrm{te}}).$$

**Observation Noise** To account for observation noise, we can augment the training covariance matrix by adding the noise covariance matrix $\Sigma$. This results in a modified covariance matrix:

$$K(X_{\mathrm{tr}}) + \Sigma.$$

For i.i.d. normal observation noise with a common variance $\sigma_{\mathrm{obs}}^2$ to the $p$ components, the noise covariance becomes a scaled identity matrix, so the training covariance matrix is given by:

$$K(X_{\mathrm{tr}}) + \sigma_{\mathrm{obs}}^2 I_{pn}. \tag{12}$$

## A.3. Training of the Gaussian Process

For a matrix-valued kernel $K_{\boldsymbol{\theta}}$ parametrised by $\boldsymbol{\theta} \in \mathbb{R}^q$, at which $K_{\boldsymbol{\theta}}$ is invertible, we tune the kernel parameters $\boldsymbol{\theta}$ using the maximum likelihood approach, i.e. minimizing the negative twice log-likelihood (n2ll) given in noise-free settings by

$$l(\boldsymbol{\theta}) = \boldsymbol{z}_{\mathrm{tr}}^\top K_{\boldsymbol{\theta}}(X_{\mathrm{tr}})^{-1} \boldsymbol{z}_{\mathrm{tr}} + \log |K_{\boldsymbol{\theta}}(X_{\mathrm{tr}})| + 2n \log 2\pi. \tag{13}$$

In our experiments, training is performed using the gradient-based Adam optimiser. The gradient of (13) with respect to $\boldsymbol{\theta}$ for values at which $K_{\boldsymbol{\theta}}(X_{\mathrm{tr}})$ is differentiable and invertible is given by

$$[2\nabla l(\boldsymbol{\theta})]_i = \boldsymbol{z}_{\mathrm{tr}}^\top K_{\boldsymbol{\theta}}^{-1}(X_{\mathrm{tr}}) \frac{\partial K_{\boldsymbol{\theta}}(X_{\mathrm{tr}})}{\partial \theta_i} K_{\boldsymbol{\theta}}^{-1}(X_{\mathrm{tr}}) \boldsymbol{z}_{\mathrm{tr}} - \mathrm{tr}\left( K_{\boldsymbol{\theta}}^{-1}(X_{\mathrm{tr}}) \frac{\partial K_{\boldsymbol{\theta}}(X_{\mathrm{tr}})}{\partial \theta_i} \right).$$

In the noisy setting (12), the n2ll and its gradient are computed analogously with the observation noise as an additional tunable kernel parameter. The parameters $\boldsymbol{\theta}$ are optimized using maximum likelihood estimation with the gradient-based Adam optimizer Kingma & Ba (2014). For experiment 5.1, the optimization is run for 1000 iterations with a learning rate of 0.01, starting from the initial values $\boldsymbol{\theta}_{\mathrm{init}} = (1, 1, 1, 1, 0.1)$.

To evaluate the predictions, we measure the average magnitude of prediction error on the test set, given by the RMSE:

$$\mathrm{RMSE} = \sqrt{\frac{1}{n_{\mathrm{te}}} \|\boldsymbol{z}_{\mathrm{te}} - \boldsymbol{m}_{\mathcal{D}_n}(X_{\mathrm{te}})\|_2^2}, \tag{14}$$

where $\boldsymbol{z}_{\mathrm{te}} \in \mathbb{R}^{pn_{\mathrm{te}}}$ is the stacked vector of observed test set responses, and $\boldsymbol{m}_{\mathcal{D}_n}(X_{\mathrm{te}})$ is the GP posterior mean at locations $X_{\mathrm{te}}$, as defined in Eq. (11) of Appendix A.

To measure the probabilistic predictive accuracy, we use the average logarithmic score LogS. For a given train/test split, the LogS is defined by the logarithmic posterior density of the test data.

### A.4. Group and representation theory

In this section, we outline the essential background on groups and group representations required for constructing equivariant kernel matrix functions. Our discussion is primarily based on the group theory framework in Reisert & Burkhardt (2007) and fundamental regions in Ginsbourger et al. (2012). For clarity and consistency, we harmonize some notations.

**Definition A.1.** *A group $(G, \circ)$ is a set $G$ equipped with a binary operation*

$$\circ : G \times G \to G, \quad (a, b) \mapsto a \circ b$$

*which satisfies*

1. *$\forall a, b, c \in G, a \circ (b \circ c) = (a \circ b) \circ c$*

2. *$\exists e \in G$ s.t. $\forall a \in G, a \circ e = e \circ a = a$*

3. *$\forall a \in G, \exists a^{-1} \in G$ s.t. $a \circ a^{-1} = a^{-1} \circ a = e$.*

**Definition A.2.** *If the set $G$ is furthermore a topological space and the group operation $\circ$ as well as its inverse $\circ^{-1}$ are continuous with respect to the topology of $G$, we call $G$ a topological group. In addition, a topological group $G$ is compact if it is compact with respect to its topology, i.e. if each open cover of $G$ admits a finite subcover.*

**Definition A.3.** *A group representation $\rho$ maps elements from a group $G$ to $L(V)$ where $V$ is a finite dimensional Hilbert space and $L(V)$ the space of linear transformations on $V$. Furthermore $\rho$ is a homomorphism, i.e. for any $g_1, g_2 \in G$, $\rho(g_1 \circ g_2) = \rho(g_1)\rho(g_2)$.*

**Remark A.4.** *There exist definitions of group representations where $V$ is infinite dimensional. However, these cases go beyond the scope of this paper, which focuses exclusively on finite-dimensional representations.*

**Definition A.5.** *A group $G$ is called a linear group if there exists an injective homomorphism $\phi : G \to GL(p, \mathbb{F})$ to the general linear group $GL(p, \mathbb{F})$ for some integer $p$ and some field $\mathbb{F}$, such that the image $\phi(G)$ is a closed set in the natural topology on $GL(p, \mathbb{F})$, which corresponds to the topology induced by the standard norm on $K^p$.*

As consequence, a linear group admits invertible matrix-valued representations.

**Definition A.6.** *A linear group representation is called unitary if for any $g \in G$ it holds $\rho_{g^{-1}} = \rho_g^\dagger$.*

**Remark A.7.** *Since our work focuses on the case $\mathbb{F} = \mathbb{R}$, a unitary representation is equivalent to an orthogonal representation, meaning that for any $g \in G$, it holds that $\rho_{g^{-1}} = \rho_g^\top$.*

Following Reisert & Burkhardt (2007), it can be shown that any representation of a finite group or of a compact continuous group with continuous representation is equivalent to a unitary representation.

**Definition A.8.** *A measure $\mu$ defined on the $\sigma-$algebra generated by the open sets of a compact group $G$, called the Borel algebra of $G$, is called left translation-invariant if for any open subset $S \subset G$ and $g \in G$ it holds $\mu(gS) = \mu(S)$, where $gS = \{g \circ s | s \in S\}$.*

By Haar's theorem, on any compact group there exists a unique left translation-invariant measure, called the left Haar measure. Analogously, there exists a unique right translation-invariant measure, called the right Haar measure.

**Definition A.9.** *A compact group is said to be unimodular, if its right and left Haar measure coincide.*

It can be shown that representations of compact, linear, unimodular groups have determinant one, which simplifies reparametrisations in Haar integrals $\int_G f(g) \, dg$.

**Definition A.10.** *A left group action of a group $G$ on a set $X$ is a mapping*

$$\Phi : G \times X \to X$$
$$(g, x) \mapsto g \star x$$

*satisfying for any $x \in X, g, h \in G$:*

1. *$e \star x = x$*

2. *$g \star (h \star x) = (g \circ h) \star x$*

**Fundamental regions**

**Definition A.11.** *The orbit of a point $x \in X$ under the action of $G$ on $X$ is the set*

$$\mathcal{O}(x) = \{g \star x \mid g \in G\}.$$

**Definition A.12.** *The stabilizer of a set $S \subset X$ in $G$ is defined by*

$$\mathrm{Stab}_\Phi(S) = \{g \in G \mid \forall x \in S, g \star x = x\}.$$

**Definition A.13.** *Let $X$ be a topological space. We call a subset $A \subset X$ a fundamental region for the action $\star$ if the following conditions hold:*

1. *$G \star \overline{A} = X$,*

2. *$(g \star A) \cap A = \emptyset \quad$ for all $g \in G \setminus \{e\}$.*

**Remark A.14.** *A fundamental region $A$ is a subset of $X$ that intersects each orbit under the action of $G$ at most once. That is, for any two distinct elements $x, y \in A$, there is no group element $g \in G$ such that $g \star x = y$.*

**Definition A.15.** *Given a fundamental region $A$, we call section any mapping $s\colon X \to G$, satisfying for all $x \in X$,*

$$s(x) \star x \in \bar{A}.$$

*We further define the associated projection map by*

$$\Pi_s\colon X \to \bar{A},$$
$$x \mapsto s(x) \star x.$$

**Remark A.16.** *By definition of $A$, the restrictions to $G \star A$ of the section $s$ and therefore also of the projection map $\Pi_s$ are uniquely defined. If the group action is free (i.e., $g \star x = x$ implies $g = e$ for all $x \in X$), then $s$ and $\Pi_s$ are unique.*

# B. More on the continuity of $K_\Pi$ in specific applications

## B.1. Continuity of $K_\Pi$ in Experiment 5.1

In Experiment 5.1, $A$ was chosen to be the positive x-axis and with the section for non-zero $x$ :

$$\rho_{s(x)} = \begin{bmatrix} \cos(\theta(x)) & -\sin(\theta(x)) \\ \sin(\theta(x)) & \cos(\theta(x)), \end{bmatrix}$$

and $s(\mathbf{0}) = I_2$. The corresponding projection map is thus $\Pi_s = (\|x\|_2, 0)$. With $\theta(x) = -\arctan(x^{(2)}/x^{(1)})$, $\rho_s$ discontinuous on $\{0\} \times \mathbb{R}$ and on $\{(-x, 0), x > 0\}$, there exists no subset $B \subset \bar{A}$ for which $\rho_s$ is continuous on $G \star B$.

However, if we use a non-angular parametrization of the representation of $s$, i.e.

$$\rho_{s(x)} = \frac{1}{\|x\|_2} \begin{bmatrix} x^{(1)} & x^{(2)} \\ -x^{(2)} & x^{(1)} \end{bmatrix} \in SO(2), \tag{15}$$

then for any subset $B \subset A$, $\rho_s$ is continuous on $G \star B$, as discontinuity occurs only at $\mathbf{0} \in \partial A$. Furthermore, $\Pi_s$ is continuous on $G \star B$ and $K_{\bar{A}}$ is continuous on $B \times B$. Thus, by Proposition 4.4, $K_\Pi$ is continuous on $(G \star A) \times (G \star A)$.

## B.2. Continuity of $K_\Pi$ and $K_\Pi^\phi$ in Experiment 5.2

For $\bar{x} = \mathrm{Vec}(\bar{a}_1, \bar{a}_2) \in \tilde{D}$, the section $s(\bar{x})$ is a rotation composed of three elementary rotations $\{\Psi_i(\bar{x})\}_{i=1}^3 \subset SO(3)$. If we parametrize the respective 2-dimensional rotation components of $\Psi_i(\bar{x})$ non-angularly as in (15), $\rho_s = \prod_{i=1}^3 \Psi_i$ is continuous except at $\bar{x} = \mathrm{Vec}(\mathbf{0}, \mathbf{0})$, which lies in the boundary of $A$. By definition of $A$, it holds for any subset $B \subset A$ that $\mathrm{Vec}(\mathbf{0}, \mathbf{0}) \notin G \star B$, hence $\rho_s$ is continuous on $G \star B$. Since $\Pi_s(\bar{x}) = \mathrm{Vec}(\rho_{s(\bar{x})}\bar{a}_1, \rho_{s(\bar{x})}\bar{a}_2)$, $\Pi_s$ is continuous on $G \star B$ for any $B \subset A$. As $K_{\bar{A}}$ is continuous on $\bar{A} \times \bar{A}$, it follows from Proposition 4.4 that $K_\Pi$ is continuous on $(G \star A) \times (G \star A)$.

Now, $\phi$ may be discontinuous on the subset of $D$ of Lebesgue measure zero given by $\mathcal{E}_\phi = \{x \in D \mid \phi(x) \in \Pi_s^{-1}(\partial A)\}$. Thus, $K_\Pi^\phi$ is continuous on $(D \setminus \mathcal{E}_\Phi) \times (D \setminus \mathcal{E}_\Phi)$.

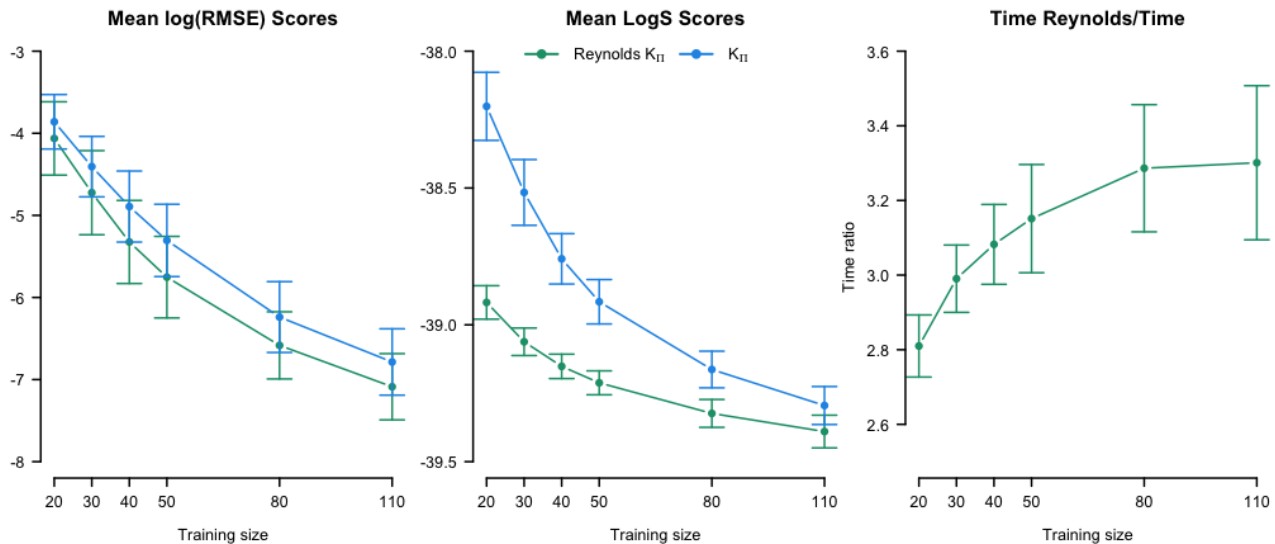

*Figure 9.* Comparison of $K_{\mathbf{\Pi}}^R$ and the original $K_{\mathbf{\Pi}}$ by the predictive scores and the ratio of computation times (in seconds).

**Extending continuity using the Reynolds operator** We can use the Reynolds operator with respect to $\star_3$, to construct a fundamental region kernel which is discontinuous only on a subset of $\mathcal{E}_\phi \times \mathcal{E}_\phi$. For this, denote by $K_{\mathbf{\Pi}}^{\mathbf{\Delta}}(\cdot,\cdot) = K_{\mathbf{\Pi}}(\mathbf{\Delta}(\cdot), \mathbf{\Delta}(\cdot))$ the $\star_2$-equivariant and $\star_1$-invariant kernel obtained by omitting the permutation of $\bar{\boldsymbol{a}}_1 = \boldsymbol{a}_2 - \boldsymbol{a}_1$ and $\bar{\boldsymbol{a}}_2 = \boldsymbol{a}_3 - \boldsymbol{a}_1$ in the construction of $\phi$. We can ensure invariance under $\star_3$ by applying the Reynolds operator for $G = \mathbb{Z}/2\mathbb{Z}$ on $K_{\mathbf{\Pi}}^{\mathbf{\Delta}}$, resulting in the $\star_{1,3}$-invariant, $\star_2$-equivariant kernel $K_{\mathbf{\Pi}}^{R\circ\mathbf{\Delta}}$ given by

$$K_{\mathbf{\Pi}}^{R\circ\mathbf{\Delta}}(\boldsymbol{x}, \boldsymbol{x}') = \frac{1}{4} \sum_{g,g' \in G} K_{\mathbf{\Pi}}^{\mathbf{\Delta}}(g \star_3 \boldsymbol{x}, g' \star_3 \boldsymbol{x}'). \tag{16}$$

Without permuting $\bar{\boldsymbol{a}}_1$ and $\bar{\boldsymbol{a}}_2$, $\mathbf{\Pi}_s$ maps onto the fundamental region

$$A^* = \left\{ \left( 0, \underbrace{\bar{a}_{12}}_{>0}, 0, \underbrace{\bar{a}_{21}}_{>0}, \bar{a}_{22}, 0 \right) \in \tilde{D} : \bar{a}_{21}, \bar{a}_{12} > 0 \right\},$$

with the same section $s$. The points of potential discontinuities of $K_{\mathbf{\Pi}}^{\mathbf{\Delta}}$ thus reduce to $\mathcal{E}_{\mathbf{\Delta}} \times \mathcal{E}_{\mathbf{\Delta}}$, where $\mathcal{E}_{\mathbf{\Delta}} = \{\boldsymbol{x} \in D \mid \mathbf{\Delta}(\boldsymbol{x}) \in \mathbf{\Pi}_s^{-1}(\partial A^*)\} \subset \mathcal{E}_\phi$. As the employed Reynolds operator may not introduce additional discontinuities, it follows that $K_{\mathbf{\Pi}}^{\mathbf{\Delta}}$ and $K_{\mathbf{\Pi}}^{R\circ\mathbf{\Delta}}$ are both continuous on $(D \setminus \mathcal{E}_{\mathbf{\Delta}}) \times (D \setminus \mathcal{E}_{\mathbf{\Delta}})$.

Comparing the GPs $GP(0, K_{\mathbf{\Pi}}^{R\circ\mathbf{\Delta}})$ and $GP(0, K_{\mathbf{\Pi}}^\phi)$ by their learning curves in Figure 9, we see that continuous $K_{\mathbf{\Pi}}^{R\circ\mathbf{\Delta}}$ provides improved predictive performance, requiring a moderate multiple of $K_{\mathbf{\Pi}}^\phi$'s computation time.

## C. Proofs

### C.1. Proof of Proposition 4.1

**Proposition 4.1** Let $G$ be a linear group acting on $D$ via $\star$, possessing a unitary group representation $\rho : g \in G \to \rho_g \in \mathbb{R}^{p \times p}$, and let $A \subset D$ be a fundamental region of $\star$. Then, for any matrix-valued kernel $K_{\bar{A}}$ on $\bar{A} \times \bar{A}$, section $s$ and associated projection $\mathbf{\Pi}_s$, $K_{\mathbf{\Pi}}$ below defines a matrix-valued kernel equivariant (w.r.t $\star$ and $\rho$) on $(G \star A) \times (G \star A)$:

$$K_{\mathbf{\Pi}}(\boldsymbol{x}, \boldsymbol{x}') = \rho_{s(\boldsymbol{x})}^\top K_{\bar{A}}(\mathbf{\Pi}_s(\boldsymbol{x}), \mathbf{\Pi}_s(\boldsymbol{x}'))\rho_{s(\boldsymbol{x}')}.$$

**Proof.** $K_{\mathbf{\Pi}}(\boldsymbol{x}, \boldsymbol{x}') = K_{\mathbf{\Pi}}(\boldsymbol{x}', \boldsymbol{x})^\top$ holds for any $\boldsymbol{x}, \boldsymbol{x}' \in D$, and positive semi-definiteness follows directly as for any $n \geq 1$, $\boldsymbol{a}_1, \ldots, \boldsymbol{a}_n \in \mathbb{R}^p$, and $\boldsymbol{x}_1, \ldots, \boldsymbol{x}_n \in D$,

$$
\begin{aligned}
\sum_{1 \leq i,j \leq n} \boldsymbol{a}_i^\top K_{\mathbf{\Pi}}(\boldsymbol{x_i}, \boldsymbol{x_j}) \boldsymbol{a}_j &= \sum_{1 \leq i,j \leq n} \boldsymbol{a}_i^\top \rho_{s(\boldsymbol{x_i})}^\top K_{\bar{A}}(\mathbf{\Pi}_s(\boldsymbol{x_i}), \mathbf{\Pi}_s(\boldsymbol{x_j})) \rho_{s(\boldsymbol{x_j})} \boldsymbol{a}_j \\
&= \sum_{1 \leq i,j \leq n} \boldsymbol{b}_i^\top K_{\bar{A}}(\mathbf{\Pi}_s(\boldsymbol{x_i}), \mathbf{\Pi}_s(\boldsymbol{x_j})) \boldsymbol{b}_j \\
&\geq 0,
\end{aligned}
$$

where $\boldsymbol{b}_i := \rho_{s(\boldsymbol{x_i})} \boldsymbol{a}_i$ and the last inequality follows from positive definiteness of $K_{\bar{A}}$.

Now, let $(\boldsymbol{x}, \boldsymbol{x}') \in (G \star A) \times (G \star A)$. Since the projector $\mathbf{\Pi}_s$ is constant on the orbits of $\boldsymbol{x}$ and $\boldsymbol{x}'$, it holds for any $g, h \in G$ :

$$
\begin{aligned}
K_{\mathbf{\Pi}}(g \star \boldsymbol{x}, h \star \boldsymbol{x}') &= \rho_{s(g \star \boldsymbol{x})}^\top K_{\bar{A}}(\mathbf{\Pi}_s(g \star \boldsymbol{x}), \mathbf{\Pi}_s(h \star \boldsymbol{x}')) \rho_{s(h \star \boldsymbol{x}')} \\
&= \rho_{s(g \star \boldsymbol{x})}^\top K_{\bar{A}}(\mathbf{\Pi}_s(\boldsymbol{x}), \mathbf{\Pi}_s(\boldsymbol{x}')) \rho_{s(h \star \boldsymbol{x}')}.
\end{aligned}
$$

Its straightforward to see that $s(g \star \boldsymbol{x}) = s(\boldsymbol{x}) \circ g^{-1}$ and thus $\rho_{s(g \star \boldsymbol{x})} = \rho_{s(\boldsymbol{x})} \rho_{g^{-1}}$. Hence,

$$
\begin{aligned}
K_{\mathbf{\Pi}}(g \star \boldsymbol{x}, h \star \boldsymbol{x}') &= \rho_g \rho_{s(\boldsymbol{x})}^\top K_{\bar{A}}(\mathbf{\Pi}_s(\boldsymbol{x}), \mathbf{\Pi}_s(\boldsymbol{x}')) \rho_{s(\boldsymbol{x}')} \rho_h^\top \\
&= \rho_g K_{\bar{A}}(\boldsymbol{x}, \boldsymbol{x}') \rho_h^\top.
\end{aligned}
$$

$\square$

### C.2. Proof of Corollary 5.1

**Corollary 5.1** Assume a Gaussian random field $Z$ and a group $G$ satisfy the assumptions of Theorem 3.1, with the kernel of $Z$ being equivariant (2). Then, given any observed realization $\boldsymbol{z}_{\mathrm{tr}}$ of $\boldsymbol{Z}_{\mathrm{tr}}$ (whereby the notation of Appendix A is used), the resulting posterior distribution retains stochastically equivariant, i.e.,

$$
\forall \boldsymbol{x} \in D,\ g \in G, \quad \mathbb{P}(\boldsymbol{Z}_{g \star \boldsymbol{x}} = \rho_g \boldsymbol{Z}_{\boldsymbol{x}} \mid \boldsymbol{Z}_{\mathrm{tr}} = \boldsymbol{z}_{\mathrm{tr}}) = 1.
$$

**Proof.** For simplicity we assume the noiseless setting, as the case of present observation noise is analogous. For a test point $\boldsymbol{x} \in D$ and the training set $X_{\mathrm{tr}}$, we have a cross-covariance matrix of the form

$$
K(\boldsymbol{x}, X_{\mathrm{tr}}) = [K(\boldsymbol{x}, \boldsymbol{x_1}), \ldots, K(\boldsymbol{x}, \boldsymbol{x_n})] \in \mathbb{R}^{p \times pn}.
$$

Since $K$ satisfies the equivariance (2), it follows that for $g \in G$

$$
K(g \star \boldsymbol{x}, X_{\mathrm{tr}}) = [\rho_g K(\boldsymbol{x}, \boldsymbol{x_1}), \ldots, \rho_g K(\boldsymbol{x}, \boldsymbol{x_n})],
$$

and therefore it holds the equivariance of the posterior mean

$$
\boldsymbol{m}_{\mathcal{D}^n}(g \star \boldsymbol{x}) = \rho_g \boldsymbol{m}_{\mathcal{D}^n}(\boldsymbol{x}).
$$

Furthermore the equivariance (2) of $K$ transfers to the posterior covariance $K_{\mathcal{D}^n}$. Let $\boldsymbol{x}, \boldsymbol{x} \in D$, and $g, h \in G$, then

$$
K_{\mathcal{D}^n}(g \star \boldsymbol{x}, h \star \boldsymbol{x}) = \rho_g K_{\mathcal{D}^n}(\boldsymbol{x}, \boldsymbol{x}) \rho_h^\top.
$$

Hence, analogously to the proof of Theorem 3.1, it follows that

$$
\mathrm{Cov}(\boldsymbol{Z}_{g \star \boldsymbol{x}} - \rho_g \boldsymbol{Z}_{\boldsymbol{x}}, \boldsymbol{Z}_{g \star \boldsymbol{x}} - \rho_g \boldsymbol{Z}_{\boldsymbol{x}} \mid \boldsymbol{Z}_{\mathrm{tr}} = \boldsymbol{z}_{\mathrm{tr}}) = 0.
$$

$\square$

# D. Experiment 5.3 continued

We extend Experiment 5.3 to the case where $F^E \sim GP(0, K^E)$, is a realisation of an equivariant Gaussian process, i.e. $K^E \in \mathcal{I}$. Under the assumption that the ocean drifter velocities $\boldsymbol{F}^{\text{ref}}$ are a realisation of a second-order centred Gaussian process, we can study the ability of combinations of GPs with non-equivariant and equivariant kernels to recover the mixture parameter $\alpha$ as a parameter estimation problem.

To analyze the parameter estimation of $\alpha$ with a combination of the squared-exponential kernel $K$ and the equivariant squared exponential kernel $K_{\boldsymbol{\Pi}}$ (resulting in GP 2 below), we generate realisations of

$$\boldsymbol{F} = \alpha \boldsymbol{F}^{\text{ref}} + (1 - \alpha) GP(0, K_{\boldsymbol{\Pi}}). \tag{17}$$

Analogously, we consider the case of the equivariant part coming from the fully equivariant Helmholtz GP with $K_{\text{H}}^{\boldsymbol{\Pi}}$ defined as in Experiment 5.3:

$$\boldsymbol{F}_H = \alpha \boldsymbol{F}^{\text{ref}} + (1 - \alpha) GP(0, K_{\text{H}}^{\boldsymbol{\Pi}}). \tag{18}$$

Figure 10 presents realisations of the random vector field (18) for different values of $\alpha$. This experiment considers four Gaussian process models to evaluate the performance of single, non-equivariant kernels against combinations of equivariant and non-equivariant kernels and the ability to recover the mixture parameter $\alpha$ through an additional tunable mixture parameter $\gamma \in [0, 1]$ of such combinations:

1. $GP\left(0, K_{\text{SE}}\right)$,
2. $GP\left(0, \gamma^2 K_{\text{SE}} + (1 - \gamma)^2 K_{\boldsymbol{\Pi}}\right)$,
3. $GP\left(0, K_{\text{H}}\right)$, and,
4. $GP\left(0, \gamma^2 K_{\text{H}} + (1 - \gamma)^2 K_{\text{H}}^{\boldsymbol{\Pi}}\right)$.

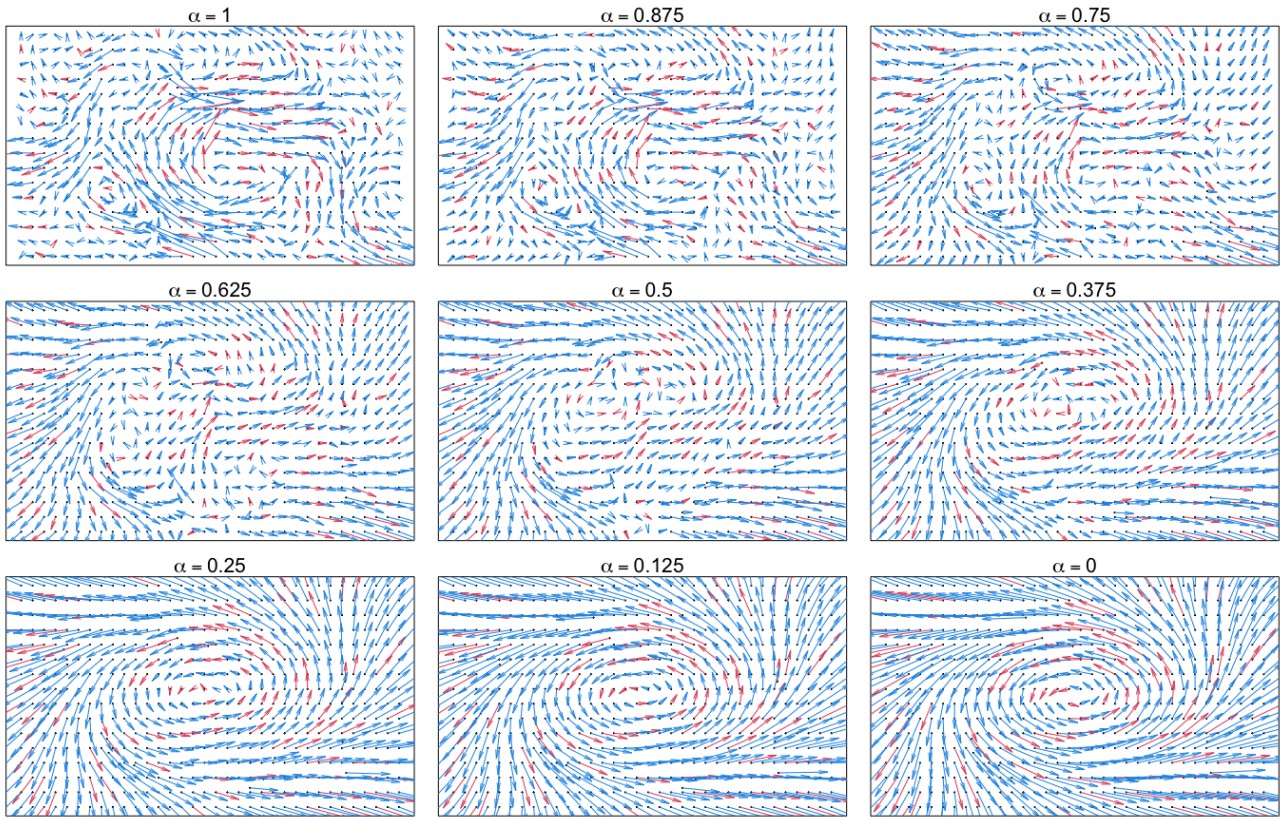

*Figure 10.* Realisations of $\boldsymbol{F}_H$ for different values of $\alpha$. Red indicates the training set of size 100.

*Table 3.* Mean performance metrics and [standard deviation]. Best scores are in bold, and values within the standard deviation of the best score given an asterisk (*).

| GP | $\alpha$ | 1 | 0.8 | 0.6 | 0.4 | 0.2 | 0 |
|---|---|---|---|---|---|---|---|
| 1 | RMSE | **0.781** [0.033] | 0.634 [0.030]* | 0.501 [0.048]* | 0.397 [0.127] | 0.306 [0.197] | 0.270 [0.275] |
|   | LogS | 0.716 [0.144]* | -0.123 [0.185]* | -1.037 [0.434] | -2.154 [1.090]* | -3.600 [2.184] | -4.273 [2.688] |
|   | Bias | - | - | - | - | - | - |
| 2 | RMSE | 0.782 [0.033]* | **0.631** [0.029] | **0.482** [0.031] | **0.334** [0.048] | **0.177** [0.048] | **0.039** [0.096] |
|   | LogS | **0.711** [0.138] | **-0.147** [0.170]* | **-1.200** [0.331] | **-2.691** [0.722] | **-5.208** [0.927] | **-6.975** [1.718] |
|   | Bias | -0.265 [0.100] | -0.124 [0.090]* | **-0.060** [0.090] | -0.027 [0.100]* | -0.018 [0.100]* | **0.014** [0.020] |
| 3 | RMSE | **0.766** [0.030] | 1.017 [1.237]* | 1.784 [3.715]* | 1.809 [3.656] | 2.278 [5.015]* | 3.416 [7.642]* |
|   | LogS | 0.863 [0.135]* | 1.184 [3.798]* | 2.126 [8.310]* | 1.083 [8.189]* | 0.270 [8.439]* | 1.029 [10.986]* |
|   | Bias | - | - | - | - | - | - |
| 4 | RMSE | 0.768 [0.031]* | **0.770** [0.721] | **1.216** [3.321] | **0.931** [2.577] | **1.174** [4.278] | **1.928** [6.900] |
|   | LogS | **0.855** [0.145] | **0.359** [1.844] | **0.166** [5.398] | **-1.584** [4.291] | **-3.511** [6.079] | **-4.407** [8.403] |
|   | Bias | **-0.139** [0.120] | **-0.054** [0.140] | -0.068 [0.130]* | **-0.022** [0.180] | **0.002** [0.140] | 0.062 [0.210] |

Table 3 summarizes the average RMSE, LogS, and bias scores across six values of $\alpha$, evaluated over 1000 samples of 100 training points and 464 test points. The kernel parameters $(\ell_1, \sigma_1, \ell_2, \sigma_2)$ of $K_{\mathbf{\Pi}}$ and $K_{\mathrm{H}}^{\mathbf{\Pi}}$ were uniformly sampled between 0 and 2. The optimization of the kernel parameters $\theta = (\ell_1, \sigma_1, \ell_2, \sigma_2, \sigma_{\mathrm{obs}}, \alpha)$ was performed using maximum likelihood estimation with the Adam optimizer over 1000 iterations and a learning rate of 0.01.

We observe that for both random vector fields $\boldsymbol{F}$ and $\boldsymbol{F}_H$, the corresponding combined GPs 2 and 4 consistently achieve lower logarithmic scores compared to the single non-equivariant kernel GPs. The combined GPs also appear to generalise better than the single non-equivariant GPs 1 and 3 in terms of point predictions, with lowest RMSE scores across all investigated values of $\alpha$ but $\alpha = 1$, where the difference in scores is insignificant. This equivalence of point predictive accuracy is explained by no additional equivariance being present in the case $\alpha = 1$.

Both combined models (GP 2 and GP 4) thus suggest strong evidence that incorporating equivariance into the model leads to better predictive performance, with the combination of non-equivariant and equivariant kernels appearing to provide a flexible framework that adapts well to varying levels of equivariance in the data.

Figure 11 presents the distribution of the estimated $\gamma$, showing that both GP 2 and GP 4 correctly identify the fully equivariant case $\alpha = 0$. However, as $\alpha$ increases, GP 2 tends to estimate the mixture parameter more accurately for $\alpha \leq 0.6$, while GP 4 performs better for $\alpha = 0.8$ and $\alpha = 1$. Increasing the number of training iterations to 3000 reduces the uncertainty in $\gamma$, as seen in the lower panel of Figure 11, confirming the benefit of extended training for more accurate parameter recovery.

This experiment demonstrates that incorporating equivariant kernels into Gaussian process models noticeably improves performance in tasks involving physical systems with inherent equivariance for better generalization and predictive accuracy.

## E. Pathological choices and their effect of fundamental regions

In Section 5.1, we discuss a shortfall of the fundamental region approach, which can occur for unfortunate constructions of the fundamental region. The left panel of Figure 12 illustrates the connected construction of $A$ from Example 4.3, for a $\mathrm{SO}(2)-$ equivariant $K_{\mathbf{\Pi}}$ taking values from $\mathbb{R}^2 \times 2\mathbb{R}^2$. The right panel shows a disconnected construction of a fundamental region using an alternating partition of the $x-$axis, which lead to reduced performance. Figure 13 shows the posterior means of two fundamental region GPs learned on samples of observations of the $\mathrm{SO}(2)-$equivariant random vector fields in Experiment 5.1. The middle column shows the posterior means of the equivariant GP with a disconnected fundamental region of 1000 sub-intervals $P_i$. The right column shows the posterior means for the connected fundamental region. In contrast to using the connected fundamental region, the posterior means of the equivariant GP with disconnected fundamental region are discontinuous and deviate significantly from the ground truths.

An analogous investigation provided equivalent conclusions for the dipole moment GPs. We constructed a disconnected fundamental region in the same way as for the $\mathrm{SO}(2)-$equivariant case, by inducing a similar alternating partition in the $\bar{x}_{22}$ component, which only requires occasional left-multiplication of the section by an additional rotation around the $y-$axis with angle $\theta = \pi$. The resulting discontinuous fundamental region GP decreases in performance, as can be seen by the elevated learning curves of Figure 14.

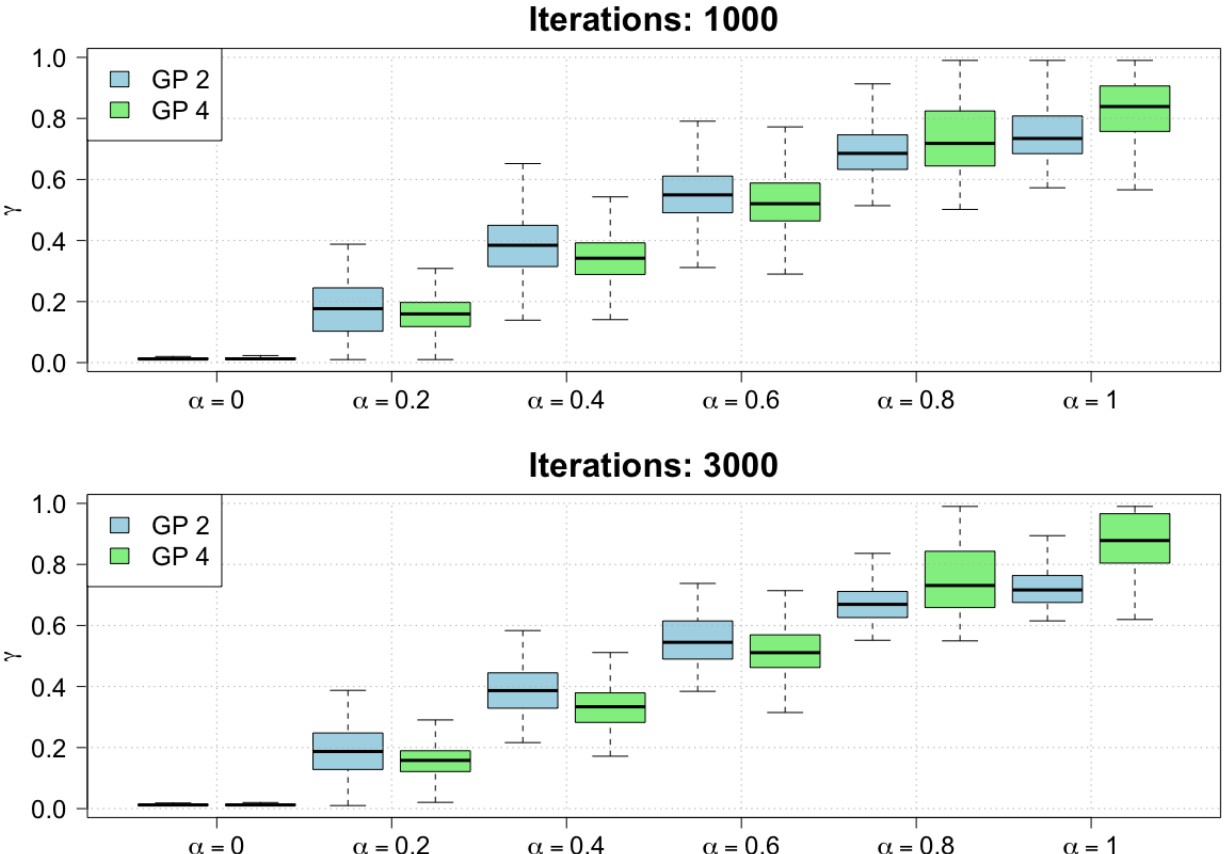

*Figure 11.* Optimised values of $\gamma$ from the mixture GPs 2 and 4 for 1000 vs. 3000 training iteration steps.

## F. An example of a matrix to vector prediction task featuring SO($d$) equivariances

We present an interesting example extending the fundamental region concept of Experiment 4.3 featuring SO($d$) equivariance in arbitrary dimension $d$. Consider a class of matrix to vector mappings constructed as follows. The Input matrices $X = [\mathbf{x}_1, \ldots, \mathbf{x}_d] \in \mathbb{R}^{d \times d}$ are chosen such that their columns form orthogonal (not necessarily orthonormal) bases of $\mathbb{R}^d$, and $f(X) \in \mathbb{R}^d$ is defined as $g(||\mathbf{x}_1||)\mathbf{x}_1$ where $g : (0, \infty) \to \mathbb{R}$. Such an $f$ is automatically equivariant with respect to SO($d$) (acting by matrix multiplication on columns of $X$ and similarly on $f(X)$) as, for any orthogonal matrix $R$,

$$f(RX) = g(||R\mathbf{x}_1||)R\mathbf{x}_1 = Rg(||\mathbf{x}_1||)\mathbf{x}_1 = Rf(X).$$

Besides, $f(X)$ does not depend on the $d-1$ last columns of $X$, which is an additional invariance property. Taking both equivariance and invariances properties into account, we arrive at the fundamental region

$$FR = \{[\alpha\mathbf{e}_1, \mathbf{e}_2, \ldots, \mathbf{e}_d], \alpha > 0\},$$

where the $\mathbf{e}_i$'s are the canonical basis vectors. The key to construct a section here is to observe that $\Psi = \left[\frac{1}{||\mathbf{x}_1||}\mathbf{x}_1, \ldots, \frac{1}{||\mathbf{x}_d||}\mathbf{x}_d\right]$ is an orthogonal matrix with

$$\Psi^T X = [||\mathbf{x}_1||\mathbf{e}_1, ||\mathbf{x}_2||\mathbf{e}_2, \ldots, ||\mathbf{x}_d||\mathbf{e}_d],$$

which can then be sent to FR by modifying the inactive columns appropriately (e.g., by norming them). Hence $f$ can be modelled via an equivariant GP model defined in terms of the latter sequence of operations and of a kernel on FR, that is, a kernel that can be parametrized on $(0, \infty) \times (0, \infty)$ (and corresponds to solely modelling the function $g$).

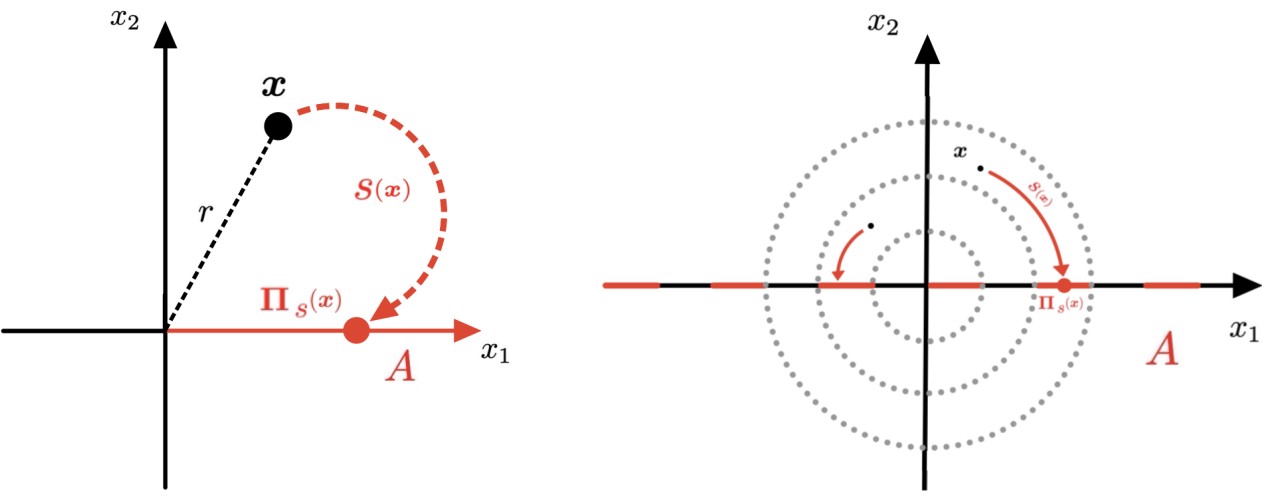

*Figure 12.* Visualisations of connected and disconnected fundamental regions for Experiment 5.1 and the associated $s$ and $\mathbf{\Pi}_s$.

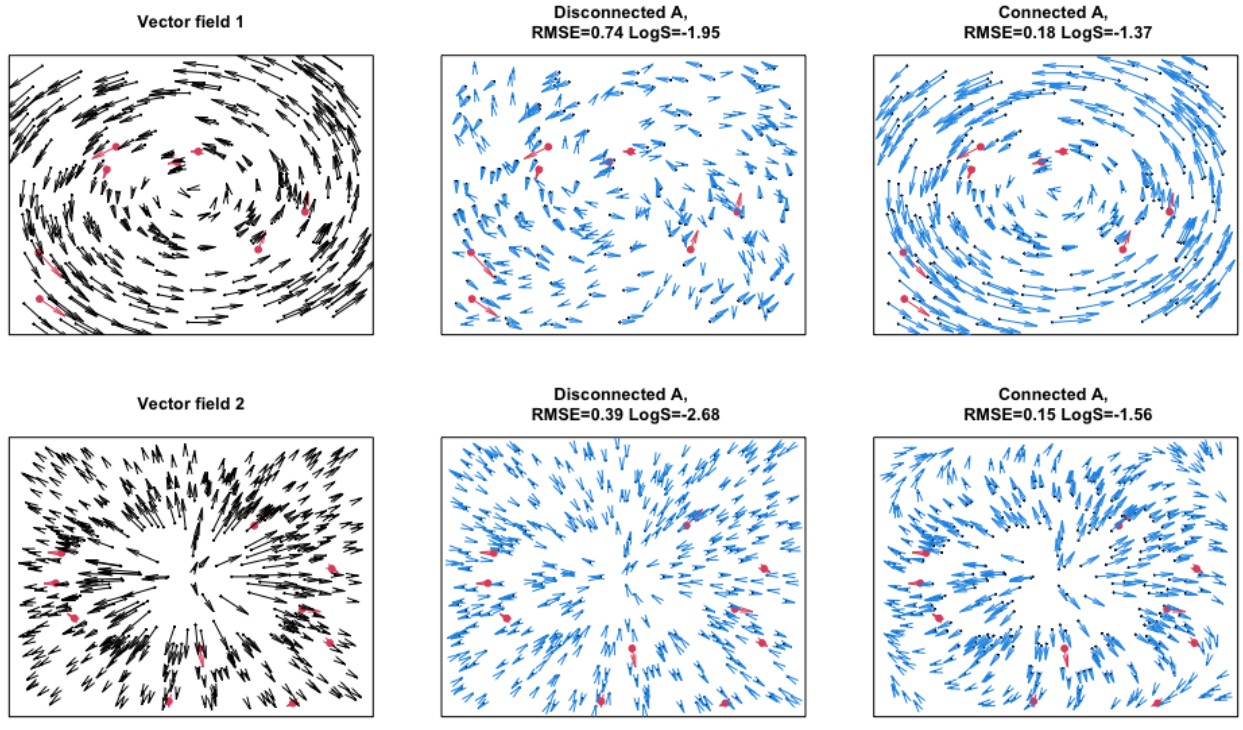

*Figure 13.* Ground truths sampled from the equivariant random fields of 5.1 (left column), posterior means of $GP(0, K_{\mathbf{\Pi}})$ with disconnected $A$ (middle column) and connected $A$ (right column).

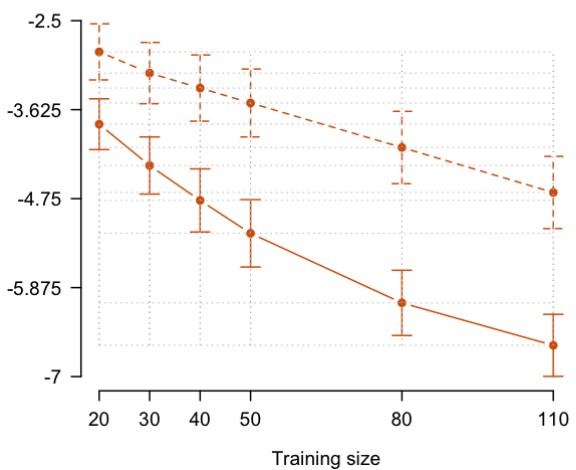
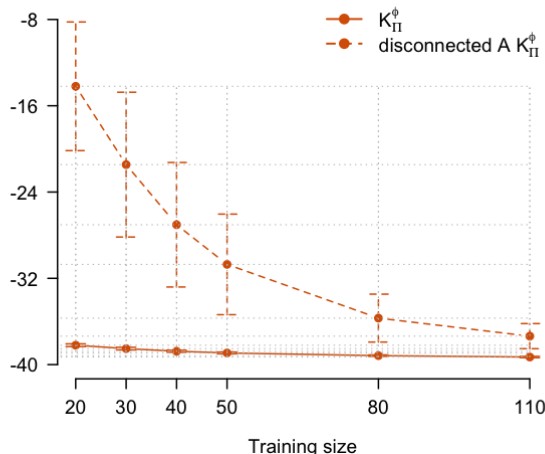

*Figure 14.* Learning curves of $GP(0, K_{\boldsymbol{\Pi}}^{\phi})$ (dotted: disconnected fundamental region) for the dipole moment prediction task.

## G. Equivariance of sparse GPs

To broaden the applicability of equivariant GP modeling to large datasets like our N-Methylformamide dataset, we introduce a (centered) sparse Gaussian Process based on $m << n$ inducing locations $X_u \in \mathbb{R}^{m \times d}$, with posterior mean and covariance respectively denoted $\boldsymbol{m}_{\mathcal{D}^n}^u$ and $K_{\mathcal{D}^n}^u$ with, for $\boldsymbol{x}, \boldsymbol{x}' \in D$,

$$\boldsymbol{m}_{\mathcal{D}^n}^u(\boldsymbol{x}) = K(\boldsymbol{x}, X_u)K(X_u)^{-1}\boldsymbol{m}_{\mathcal{D}^{n-m}}(X_u),$$
$$K_{\mathcal{D}^n}^u(\boldsymbol{x}, \boldsymbol{x}') = K(\boldsymbol{x}, \boldsymbol{x}') - K(\boldsymbol{x}, X_u)K(X_u)^{-1}(K(X_u) - K_{\mathcal{D}^{n-m}}(X_u))K(X_u)^{-1}K(X_u, \boldsymbol{x}').$$

Here, $\boldsymbol{m}_{\mathcal{D}^{n-m}}(X_u)$ and $K_{\mathcal{D}^{n-m}}(X_u)$ are the posterior mean and covariance at the inducing points given the remaining observations $\mathcal{D}^{n-m}$, given by

$$\boldsymbol{m}_{\mathcal{D}^{n-m}}(X_u) = K(X_u, X_{tr})K(X_{tr})^{-1}\boldsymbol{z}_{tr},$$
$$K_{\mathcal{D}^{n-m}}(X_u) = K(X_u) - K(X_u, X_{tr})K(X_{tr})^{-1}K(X_{tr}, X_u).$$

If this sparse GP is built upon an equivariant kernel, it will be equivariant in its mean and covariance and therefore enjoy stochastic equivariance. To see this, consider any $g, h \in G$. It holds

$$\boldsymbol{m}_{\mathcal{D}^n}^u(g \star \boldsymbol{x}) = K(g \star \boldsymbol{x}, X_u)K(X_u)^{-1}(\boldsymbol{m}_{\mathcal{D}^{n-m}}(X_u)) = \rho_g K(\boldsymbol{x}, X_u)K(X_u)^{-1}\boldsymbol{m}_{\mathcal{D}^{n-m}}(X_u))\rho_g \boldsymbol{m}_{\mathcal{D}^n}^u(\boldsymbol{x}),$$

and

$$\begin{aligned}
K_{\mathcal{D}^n}^u(g \star \boldsymbol{x}, h \star \boldsymbol{x}') &= K(g \star \boldsymbol{x}, h \star \boldsymbol{x}') - K(g \star \boldsymbol{x}, X_u)K(X_u)^{-1}(K(X_u) - K_{\mathcal{D}^{n-m}}(X_u))K(X_u)^{-1}K(X_u, h \star \boldsymbol{x}') \\
&= \rho_g K(\boldsymbol{x}, \boldsymbol{x}')\rho_h^{\top} - \rho_g K(\boldsymbol{x}, X_u)K(X_u)^{-1}(K(X_u) - K_{\mathcal{D}^{n-m}}(X_u))K(X_u)^{-1}K(X_u, \boldsymbol{x}')\rho_h^{\top} \\
&= \rho_g K_{\mathcal{D}^n}^u(\boldsymbol{x}, \boldsymbol{x}')\rho_h^{\top}.
\end{aligned}$$

It is worth noting that additional assumptions for computational efficiency like FITC and PITC will not affect stochastic equivariance of the posterior distribution of the sparse GP, as these assumptions only replace $\boldsymbol{m}_{\mathcal{D}^{n-m}}(X_u)$ and $K_{\mathcal{D}^{n-m}}(X_u)$ with approximations thereof. Similarly, conditioning on a finite number of derivatives or linear forms (e.g., Fourier coefficients) will preserve stochastic equivariance.

# H. Data generation

The dipole moments were generated by optimising an initial guess for the geometry of water using the RI-MP2 module in the TURBOMOLE quantum chemistry program. The theory for this module is described in the original MP2 paper by Møller & Plesset (1934) and the RI-MP2 paper by Weigend & Häser (1997). Furthermore, a cc-pVDZ basis set was used, as created by Dunning (1989). Using this optimised structure, a numerical hessian was found. Diagonalising the mass weighted Hessian then gave a set of three Normal coordinates, corresponding to bending, symmetric and asymmetric bond stretch. Using the static grid method Toffoli et al. (2007) in the MIDASCPP program package a grid of twenty points was constructed along each normal coordinate. These points were linearly spaced between the classical turning points of the tenth excited state of the quantum mechanical oscillator approximation which can be constructed from the mass weighted hessian. Furthermore, three twenty by twenty grids were constructed describing displacements along two out of the three coordinates simultaneously. Combined with the optimised structure this gives 1261 geometries. For each of these geometries, the electric dipole moment was calculated using the RI-MP2 method and a cc-pVDZ basis set in TURBOMOLE. To ensure dataset efficiency, 410 pairs of points that were rotations of each other were identified and removed. The remaining 851 points were then randomly rotated to cover a larger region of Cartesian space. This approach samples the geometries of chemical interest around the optimised structure and them rotates them in space resulting in a diverse and representative dataset.

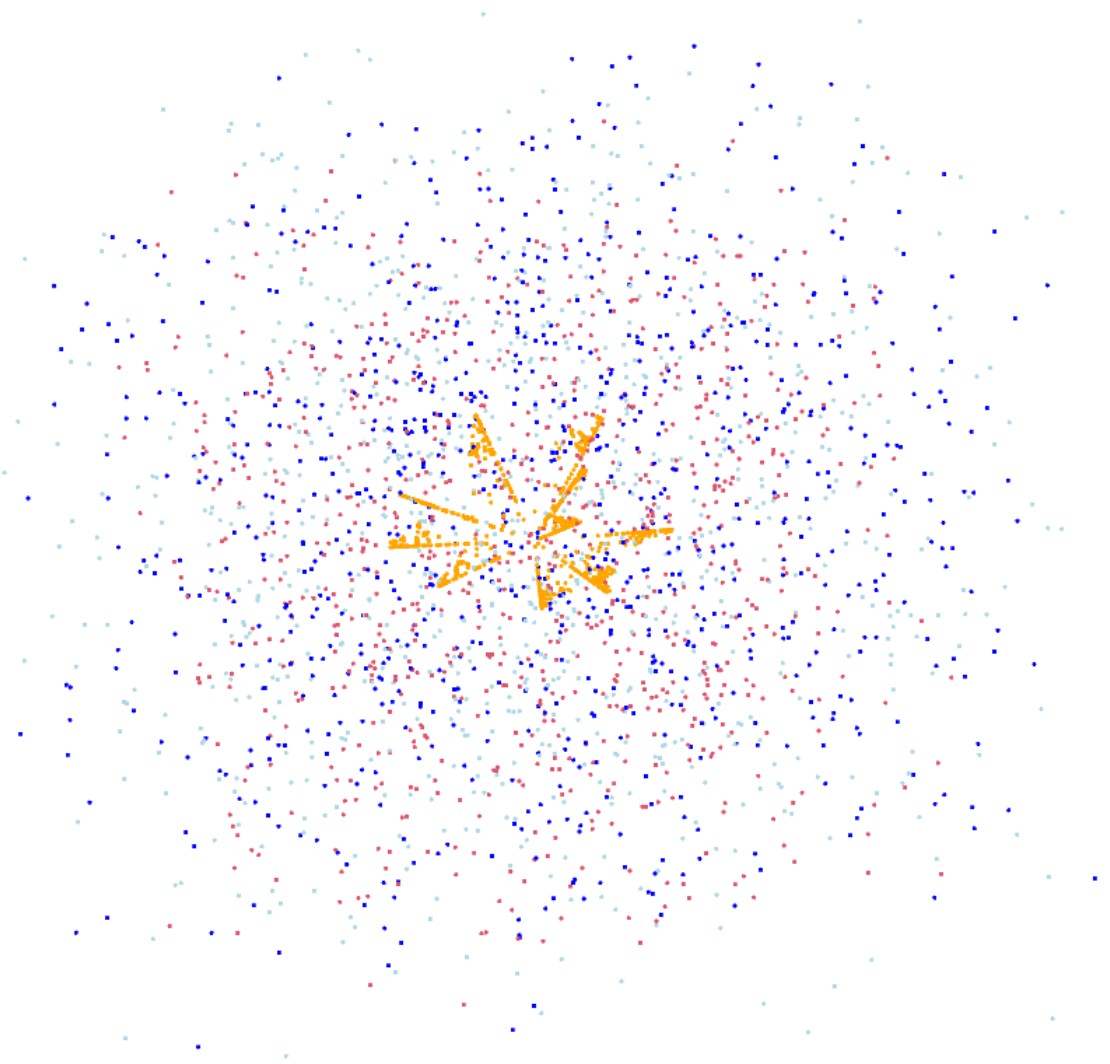

*Figure 15.* The point cloud representing the dataset: (light) blue indicate positions of the hydrogen atoms, red the position of the oxygen atoms and orange the dipole moments.

## I. Optimized kernel parameter values

We train the GPs using Adam optimisation over 1000 iterations, with all kernel parameters initialised to 1, maximising the training likelihood. Figure 16 shows the distributions of the optimised values (after initialised by 1) of the parameters $\ell, \sigma$ for 2000 draws of the training and test split for different training sizes.

The choice of starting kernel parameter values plays an influential role, the left panel of Figure 17 shows the distribution of the optimal parameter values for $GP(0, K_1)$ and $GP(0, K_{\boldsymbol{\Pi}})$ on training sets of size 50, when initialised uniformly on $[10^{-4}, 5]$. We see that $GP(0, K_1)$ tends to increase the lengthscale and decrease the variance, which indicates the kernel to smooth out the function excessively, likely because it lacks the capacity to capture the complexity in the data, which is not the case for $GP(0, K_{\boldsymbol{\Pi}})$. The right panel shows the distributions of the corresponding log(RMSE) and LogS scores after optimising the kernel parameters, indicating stable predictive performance for both GPs with respect to initialisation.

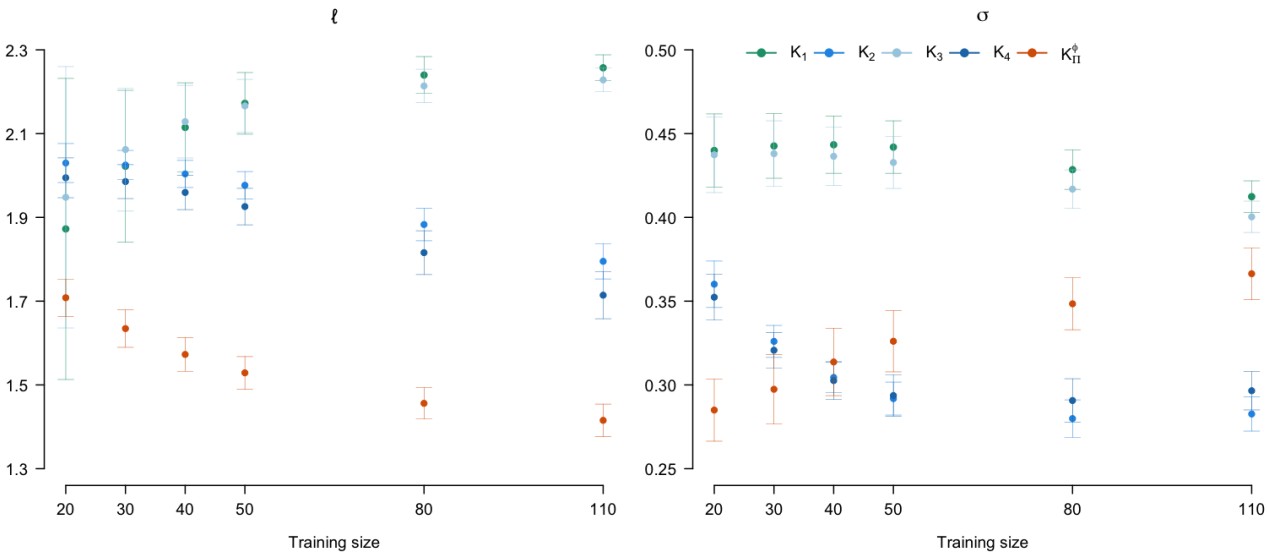

*Figure 16.* Optimised parameters (mean $\pm$ sd over 2000 samples) for different training sizes for the dipole moment prediction in 5.2.

## J. Dipole Moment Prediction for the N-Methylformamide Molecule

We consider a dataset comprising 21,000 distinct configurations of the N-Methylformamide (NMF) molecule, which consists of nine atoms. As outlined in Experiment 5.2, constructing a kernel $K_{\boldsymbol{\Pi}}$ that is rotation-equivariant and translation-invariant follows the same principles as in the water molecule setting.

Figure 18 shows preliminary learning curves comparing the baseline Gaussian process $GP(0, K_1)$ using a squared exponential kernel with the $\star_2$-equivariant and $\star_1$-invariant Gaussian process $GP(0, K_{\boldsymbol{\Pi}})$. Evaluation was conducted on 500 test points across 1,000 random train-test splits. The kernel parameters were optimized via the same Adam routine used in the water molecule experiments. For additional comparison, we also report performance using fixed kernel hyperparameters $(\sigma, \ell) = (1, 1)$ (dashed lines), which notably reduced performance—particularly for $GP(0, K_{\boldsymbol{\Pi}})$ across both metrics and for $GP(0, K)$ in terms of the log score.

The relatively flat learning curves of the baseline GP highlight its inability to model the structured nature of the data. In contrast, the equivariant GP demonstrates an improved ability to capture the structural of the data. However, achieving high predictive accuracy requires larger training sets, motivating the use of sparse GP methods, which is part of ongoing work.

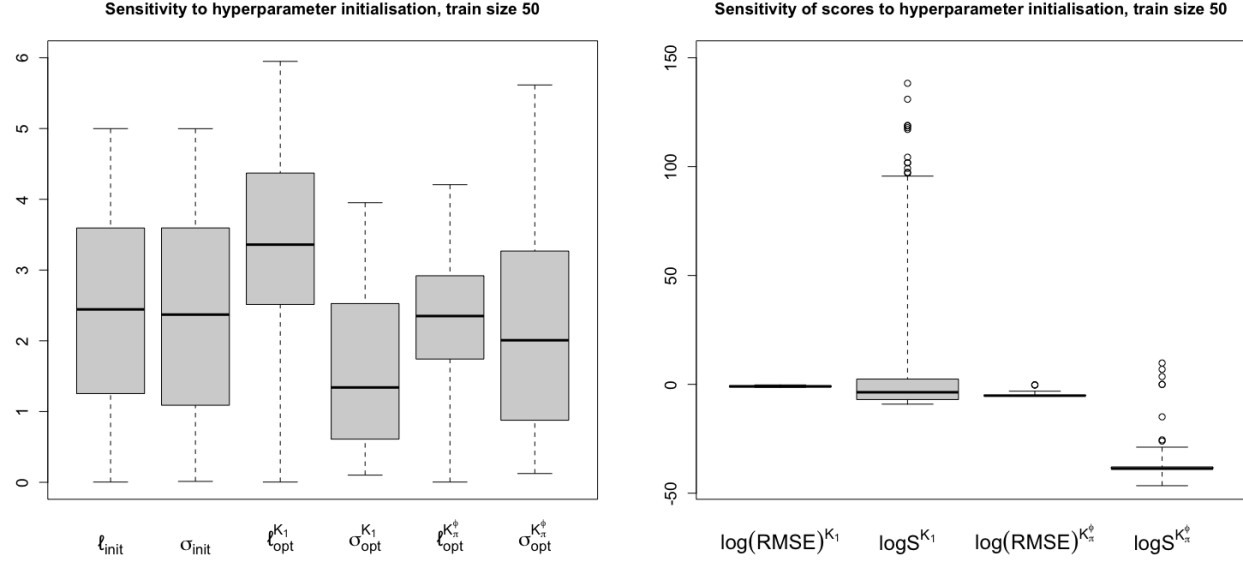

*Figure 17.* Left panel: Distributions of initial kernel parameters and the corresponding optimised values. Right panel: distribution of log(RMSE) and LogS scores after optimisation.

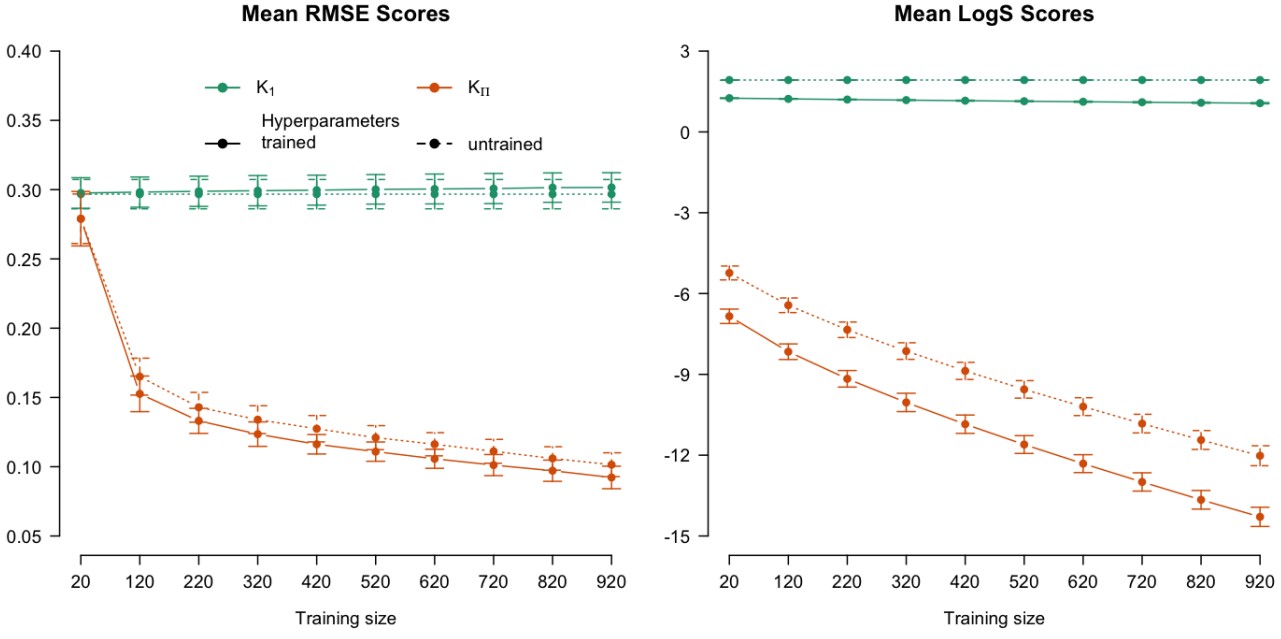

*Figure 18.* Predictive performance (mean $\pm$ std over $10^3$ repetitions) of GP models versus training set size on the N-Methylformamide molecule dataset.

