# OpenReview forum: "Integration-free Kernels for Equivariant Gaussian Process Modelling"
_ICML.cc/2025/Conference — ICML 2025 poster_

### Official Review · Reviewer_iBFz · 2025-03-11

**Overall Recommendation:** 4

**Summary:**

This paper introduces a novel class of integration-free equivariant kernels for Gaussian processes (GPs), addressing the computational inefficiency of traditional equivariant kernels that require group integrations. The key idea leverages fundamental regions to project inputs into a representative subset, enabling equivariant kernel construction without integration. Empirical validation includes molecular dipole moment prediction and ocean velocity data. The proposed kernels achieve up to 500× speedup over integration-based methods while maintaining or improving predictive accuracy (RMSE, LogS), demonstrating practical utility in scientific applications.

**Claims And Evidence:**

Computational efficiency: Figure 3 and Section 4.3 show a 45-hour vs. 55-second runtime comparison for integration-based vs. integration-free kernels validated on synthetic data.

Equivariance guarantees: Theorem 3.1 and Corollary 5.1 link kernel design to stochastic equivariance, with posterior samples in Figure 4 confirming equivariant realizations.

**Essential References Not Discussed:**

NA

**Experimental Designs Or Analyses:**

NA

**Methods And Evaluation Criteria:**

Evaluation: RMSE and LogS are appropriate metrics for regression and probabilistic calibration. Baselines (e.g., Helmholtz kernel, double-integration kernels) are well motivated.

**Other Comments Or Suggestions:**

NA

**Other Strengths And Weaknesses:**

NA

**Questions For Authors:**

1. Would adaptive section selection (e.g., optimizing A) improve robustness? I liken this paper to introducing inducing points in sparse GPs, where the positions of the inducing points can be optimized. Therefore, I wonder if the foundation region A could also be optimized.

**Relation To Broader Scientific Literature:**

If I understand correctly, the work builds on integration-based equivariant kernels and scalar invariance via fundamental regions. It extends these ideas to matrix-valued kernels and stochastic equivariance. I'm not sure if this idea could be borrowed into the sparse GP field.

**Theoretical Claims:**

I didn't check the proofs because I'm not an expert in this field.

---

> ### Author Rebuttal · Authors · 2025-03-31
>
> We would like to thank the referee very much for taking the time reviewing our work, highlighting the underlying rationale and the obtained numerical benefits within a probabilistic prediction and evaluation framework, and pointing out further aspects that will deepen our work. Two specific directions stand out, and we think that they are both very valuable for our work and follow-ups thereof: scaling up the approach via sparse GP modelling, and criteria to select/update A.
>
> The referee's point on sparse GPs inspired us to extend our theoretical framework, for which we below establish how stochastic equivariance of a GP guarantees stochastic equivariance of a sparse version of it. This result may broaden the applicability of equivariant GP modeling to large datasets, like with our new large N-Methylformamide molecule dipole moment dataset (See first results at https://equivariantrf.github.io/Equivariant-Random-Fields/reviewer-discussion-html.html). While our preliminary results with training/test sets of moderate size are very promising indeed, we envision on a longer run to apply sparse GP modelling to the full data set comprising 20000 molecules of each 9 atoms (that would ideally be benchmarked against equivariant neural networks following up on referee HxJv's suggestions).
>
> The gist of the extension of equivariance properties to sparse GPs is to build upon Corollarly 5.1. Let us explain that next.
> A (centered) sparse GP based on $m<<n$ inducing locations $X_u \\in \\mathbb{R}^{m\\times d}$ possesses the following posterior distribution in terms of the inducing locations
> $Z\\mid \\mathcal{D}^n \\sim \\mathcal{N}(m_{\\mathcal{D}^n}^u, K_{\\mathcal{D}^n}^u),$ where for $\\boldsymbol{x},\\boldsymbol{x'}\\in D$
>
> \\begin{equation*}
>     m_{\\mathcal{D}^n}^u(\\boldsymbol{x})=K(\\boldsymbol{x},X_u)K(X_u)^{-1}m_{\\mathcal{D}^{n-m}}(X_u)
> \\end{equation*}
>
> and
> \\begin{equation*}
> K_{\\mathcal{D}^n}^u(\\boldsymbol{x},\\boldsymbol{x'})=K(\\boldsymbol{x},\\boldsymbol{x'})-K(\\boldsymbol{x},X_u)K(X_u)^{-1}(K(X_u)-K_{\\mathcal{D}^{n-m}}(X_u))K(X_u)^{-1}K(X_u,\\boldsymbol{x'}).
> \\end{equation*}
> Here, $m_{\\mathcal{D}^{n-m}}(X_u)$ and$ K_{\\mathcal{D}^{n-m}}(X_u)$ are the posterior mean and covariance of the field at the inducing points given the remaining observations $\\mathcal{D}^{n-m},$ given by
> $$
> m_{\mathcal{D}^{n-m}}(X_u) = K(X_u,X_{tr})K(X_{tr})^{-1}\boldsymbol{z}_{tr}
> $$
>
> and
>
> $$
> K_{\mathcal{D}^{n-m}}(X_u) = K(X_u)-K(X_u,X_{tr})K(X_{tr})^{-1}K(X_{tr},X_u).
> $$
>
> Analogously to the proof of Corollary 5.1, the posterior distribution of the sparse version of a stochastically equivariant GP is stochastically equivariant as well, since for any $g,h \\in G,$
> \\begin{equation*}
>     m_{\\mathcal{D}^n}^u(g\\star \\boldsymbol{x})=K(g\\star \\boldsymbol{x},X_u)K(X_u)^{-1}m_{\\mathcal{D}^{n-m}}(X_u)= \\rho_g K(\\boldsymbol{x},X_u)K(X_u)^{-1}m_{\\mathcal{D}^{n-m}}(X_u)=\\rho_g m_{\\mathcal{D}^n}^u(\\boldsymbol{x}),
> \\end{equation*}
>
> \\begin{align*}
> &K_{\\mathcal{D}^n}^u(g\\star\\boldsymbol{x},h\\star\\boldsymbol{x'})\\\\
> =&K(g\\star\\boldsymbol{x},h\\star\\boldsymbol{x'})-K(g\\star\\boldsymbol{x},X_u)K(X_u)^{-1}(K(X_u)
> -K_{\\mathcal{D}^{n-m}}(X_u))K(X_u)^{-1}K(X_u,h\\star\\boldsymbol{x'})\\\\
> =&\\rho_gK(\\boldsymbol{x},\\boldsymbol{x'})\\rho_h^T-\\rho_gK(\\boldsymbol{x},X_u)K(X_u)^{-1}(K(X_u)-K_{\\mathcal{D}^{n-m}}(X_u))K(X_u)^{-1}K(X_u,\\boldsymbol{x'})\\rho_h^T\\\\
> =&\\rho_gK_{\\mathcal{D}^n}^u(\\boldsymbol{x},\\boldsymbol{x'})\\rho_h^T.
> \\end{align*}
>
> Similarly, conditioning on a finite number of derivatives or linear forms (e.g., Fourier coefficients) will preserve stochastic equivariance.
>
> Concerning the choice of $A, s,\Pi_s$, while we have no procedure to construct them in a generic case-independent fashion, our additional tests illustrate that connectedness is a desirable feature for $A$. Automatically exploring the set of possible $A, s,\Pi_s$ appear as a fascinating problem and as a daunting task.
> As of now, we are unaware of general adaptive selection methods in the employed fundamental region approach and would welcome suggestions. Let us add that the choice of $K_A$ and the interplay with $s$ and $\Pi_s$ also offers interesting degrees of freedom as we will further stress in the discussion. Our first thoughts at this stage is that one could use likelihood- or cross-validation-based approaches as a means to compare several candidates for a corresponding tuple (consisting of $A, s, $ and possibly $K_A$). This could be performed straightforwardly on a finite set of tuples, and could be extended to parametric families. But already in the two-dimensional example 4.3, there are many possible ways to perform these choices. Let us observe that our initial choice not only feature connectedness but also "flatness"; taking {(x,h(x)), x>0} with $h:(0,\infty) \to \mathbb{R}$ a continuous non-decrasing mapping such that $h(0)=0$ would work, too. We would be happy to include a perspective on that!

---

### Official Review · Reviewer_u1Nr · 2025-03-13

**Overall Recommendation:** 4

**Summary:**

This paper introduces the group-theoretic notion of fundamental regions and proposes a feasible method to construct kernels for equivariant functions. The proposed method is free of integration operations and much faster than conventional methods. Experiments on synthetic and real-world data confirmed the model's validity.

**Claims And Evidence:**

The main claim that the proposed method can efficiently construct kernels with equivariant property is supported by clear and convincing evidence.

**Essential References Not Discussed:**

Equivariance is closely related to physical theories, and similarly, symplectic Gaussian process regression [a] provides a kernel method-based approach that preserves physical properties. Please discuss the relationship between the prior study and the proposed method.

[a] Rath et al., Symplectic Gaussian process regression of maps in Hamiltonian systems. Chaos: An Interdisciplinary Journal of Nonlinear Science, 31(5):053121, 2021.

**Experimental Designs Or Analyses:**

No issues specified.

**Methods And Evaluation Criteria:**

The proposed methods and evaluation criteria make sense.

**Other Comments Or Suggestions:**

No other comments.

**Other Strengths And Weaknesses:**

Weakness
- The pros/cons of $K_{\int}$ and $K_{\pi}$ are not clear. It is clear that $K_{\pi}$ is superior to B in terms of computational efficiency, but does $K_{\pi}$ have any limitations in terms of expressiveness? Currently, the paper seems to present $K_{\pi}$ as a complete superset of $K_{\int}$, so it would be helpful to clarify this point explicitly.

**Questions For Authors:**

No other question.

**Relation To Broader Scientific Literature:**

Equivariance, along with invariance, is an important property in physics. The proposed method could provide a more accurate approach to estimating underlying functions related to physical phenomena with limited observation.

**Theoretical Claims:**

Theoretical claims seem to be correct.

---

> ### Author Rebuttal · Authors · 2025-03-31
>
> We would like to thank the referee for taking the time to review our work and the constructive comments that will truly help us improving the paper. We really appreciate the suggestion to open on other physical knowledge that can be incorporated in GP models and kernel methods. We discovered the suggested reference on symplectic GP with great interest. We found in particular that a parallel result to our kernel characterization of stochastic equivariance could be established for the relevant symplectic property, so that the kernel proposed in the symplectic GP paper could drive under suitable assumptions (regularity) not only the symplectic nature of the GP posterior mean but also of the posterior sample functions (almost surely). While this is not falling into the umbrella of equivariance, we will explain how the two could be connected via null spaces of linear operators. This connection has already been made in other contexts, for example in scalar-valued GP settings. We will open perspectives on how, under appropriate assumptions, equivariances, the symplectic property, and other properties of vector fields such as being divergence- or curl-free may be treated in a unified way (via linear constraints) when it comes to their incorporation in GP modelling. Let us remark that the fundamental region approach might not be straightforward to transport broadly beyond group invariances and equivariances.
>
> Coming to clarifications regarding comparisons between integration vs fundamental region approaches, we are happy to have the opportunity to clarify this is in the paper, as it is really not our intention to suggest that the fundamental domain approach is generally superior to integration.  We do think that integration-based equivariant kernels are very nicely theoretically grounded, as our projection result illustrates, and also may deliver higher or lower performances depending on the applications. However, we stumbled across prohibitive costs in case of larger / infinite groups, and we found fundamental region approaches to deliver a fast alternative also passing the argumentwise equivariance requirements, and doing the job on our challenging molecule application(s). It is yet very interesting to note, and we will stress this further, that combining the fundamental region approach for SO(3) with an Reynolds operator approach (with a group of order two in that case) delivered better performances than the pure fundamental region approach (See Appendix B). As mentioned also in the response to reviewer HxJv, we consider putting these results more to the fore as this subtle combination may be extended to further contexts.
>
> We also wish to thank the referee for stressing the context of “limited observation” within which our work takes its roots. GPs are known to be especially suitable in such contexts, providing a flexible family of probabilistic predictors able to work with scarce training data. Standard GP models are actually known to possess limitations when it comes to bigger data sets, and sparse GP models have been developed to extend the scalability of GPs.  Reviewer iBFz attracted our attention on that and enquired whether our results and constructs could be transposed to the field of sparse GP modelling. As we develop in the response to reviewer iBFz, we are happy to respond positively to the latter and explain how sparse GPs built upon an argumentwise equivariant kernel (and in particular on the fundamental region kernels) will have their mean and kernel inheriting that property and therefore enjoy stochastic equivariance. As illustrated on  https://equivariantrf.github.io/Equivariant-Random-Fields/reviewer-discussion-html.html and further discussed in other rebuttals, we conducted experiments highlighting how poor fundamental region choices may affect performances, and also obtained promising first results in extending the integration-free approach to bigger (9-atom) molecules.
>
> We hope that our efforts to address the various comments from this and the other reviews will be appreciated and considered to clarify what needed to be / improve the paper, and that this could justify score increases enabling our work to be part of ICML 2025.

---

> > ### Comment · Reviewer_u1Nr · 2025-04-03
> >
> > I thank the authors for the clarifications. My concerns have been addressed, and I will raise my score to 4: accept. I’m also impressed by the fact that stochastic equivariance still holds for the sparse version of GPs, which I think should be highlighted in the paper.

---

### Official Review · Reviewer_bAhY · 2025-03-13

**Overall Recommendation:** 4

**Summary:**

The paper introduces an integration-free approach to constructing equivariant kernels, leveraging the concept of fundamental domains of the action. The authors demonstrate the effectiveness of their method through one synthetic example and two real-world applications, highlighting its practicality and potential impact.

**Claims And Evidence:**

Yes, all the claims made are supported by convincing evidence.

**Essential References Not Discussed:**

N/A

**Ethical Review Flag:**

Flag this paper for an ethics review.

**Experimental Designs Or Analyses:**

N/A.

**Methods And Evaluation Criteria:**

Yes, the datasets used are relevant for the problem tackled, and the synthetic experiment is a nice visualization of the impact of using $K_{\pi}$ rather than $K_\int$.

**Other Comments Or Suggestions:**

- Please use the \citet comment when citing in text, otherwise looks odd.
- I guess $\bar{A}$ is the closure of $A$, but it's not explained in the paper.

**Other Strengths And Weaknesses:**

The paper is well written and it does a great job of breaking down complex mathematical concepts and using examples to make them easier to understand. Overall, I think it’s a solid and valuable contribution, especially in making equivariant kernels more computationally feasible, an important challenge for the community.

**Questions For Authors:**

N/A

**Relation To Broader Scientific Literature:**

The paper is a timely contribution to the literature, providing a computationally feasible approach to equivariant kernels. This is crucial for images, proteins, and graph-structured data, where symmetry plays a key role. Introducing an integration-free method offers a practical solution to a long-standing challenge in this area.

**Theoretical Claims:**

I checked the proof of Theorem 3.1, Proposition 4.1 and Corollary 5.1; they look correct to me. I didn't check the proof of Theorem 3.2.

---

> ### Author Rebuttal · Authors · 2025-03-31
>
> We warmly thank the referee for taking the time to check our results and for the very encouraging positive evaluation. We will of course fix the citation command and precise the closure notation. We hope that the referee will also appreciate the overall discussion and our efforts to further improve the paper (including new experiments with bigger molecules, stochastic equivariance carrying over to sparse GP models, and further response points that can be found in rebuttals to other referees). Please see https://equivariantrf.github.io/Equivariant-Random-Fields/reviewer-discussion-html.html
> and the other rebuttals for new illustrations and discussions on our results and extensions thereof.

---

### Official Review · Reviewer_HxJv · 2025-03-19

**Overall Recommendation:** 3

**Summary:**

This paper introduces integration-free equivariant kernels to avoid computationally expensive integration. The method is claimed to be computationally efficient while preserving equivariance. Applications in velocity fields and molecular dipole moments are used to demonstrate effectiveness. While promising, the approach remains limited to low-dimensional groups, lacks extensive theoretical guarantees on stability, and fails to provide a rigorous scalability analysis.

**Claims And Evidence:**

The main claim of achieving computational efficiency while preserving equivariance is partially substantiated. While empirical results show significant speedups, they are restricted to low-dimensional cases, and generalization to larger groups remains unclear. Some ablation studies on region choice and its impact on kernel continuity are missing.

**Essential References Not Discussed:**

N\A

**Experimental Designs Or Analyses:**

The velocity field experiments and molecular provide a useful baseline, albeit at times offers only marginal improvements. The ocean dataset experiment is interesting but limited.

**Methods And Evaluation Criteria:**

Theoretical derivations are solid but rely on restrictive assumptions which do not always hold in real-world applications. The evaluation criteria focus on RMSE and LogS. Hyperparameter selection discussion is lacking wrt sensitivity and initialization effects, potentially affecting model reliability.

**Other Comments Or Suggestions:**

More structured discussion on practical limitations would strengthen the impact of the work.

**Other Strengths And Weaknesses:**

The method is computationally efficient and theoretically justified. However, the lack of scalability analysis, absence of hyperparameter sensitivity studies, and limited exploration of high-dimensional problems weaken the overall contribution.

**Questions For Authors:**

How does the method scale to higher-dimensional equivariant problems?

How sensitive is performance to fundamental region choice?

How does it compare to equivariant neural networks in terms of sample efficiency?

**Relation To Broader Scientific Literature:**

The work builds on prior studies of equivariant kernels and methods methods.

**Theoretical Claims:**

The theoretical characterization is rigorous and a strong point of the paper.

---

> ### Author Rebuttal · Authors · 2025-03-31
>
> We warmly thank the referee for taking the time to review our work and the stimulating comments. The question and comments pertaining to scaling to higher-dimensional equivariant problems call for a distinction between group order, input dimensions, and training set size (in particular). The statement “While promising, the approach remains limited to low-dimensional groups” puzzled us at first because our examples include infinite groups (SO(2), SO(3)), so we hypothesized that the mentioned limitation could pertain, e.g., to either the dimensionality of inputs or the data set cardinality. In the submitted paper, our examples featured 2- and 9-dimensional inputs. During the rebuttal, on the molecule side, we have been able to tackle a bigger molecule (9 atoms, i.e. 27-dimensional), and we are happy to report that our first results on the applicability of kernels based on analogous fundamental regions are very promising. Cf. learning curves of GPs for N-Methylformamide molecule dipole moments (20 to 100 training molecules at this stage, 500 test ones) at
>
> https://equivariantrf.github.io/Equivariant-Random-Fields/reviewer-discussion-html.html
>
> Also, we have set up a proof of concept fundamental region example of matrix to vector prediction tasks featuring SO(d) equivariances in arbitrary dimension. It basically consists in mapping matrices with columns forming an orthogonal (but not necessary orthonormal) basis of $R^d$ to a scalar non-linear function of the norm of the first column times the first column. In such a case one may prove that ${[\alpha e_1,e_2,...,e_n], \alpha >0}$ forms a fundamental region and provide a section and projection. We may include this example in appendix.
>
> Coming to the scalability in terms of training data size (n), while standard GP implementations can be tricky to scale up (notably due to O(n^3) covariance matrix inversion costs), there are options such as sparse GPs (see discussion with referee iBFz) that allow tackling large n applications. The discussion led us to look further into the matter of extending our results and constructions to sparse GPs are we found out that our results do carry over nicely in that context, in the sense that if one uses an argumentwise equivariant kernel as original kernel in sparse GP modelling, the resulting sparse GP will retain the equivariance property in the mean and the covariance.
>
> We will stress this interesting fact and also stress in the paper’s discussion that a research perspective that expects us beyond this work is to scale it up via sparse GPs and benchmark the resulting sparse equivariant GPs against equivariant neural networks (sending us to the referee’s third question). On the latter point, we are very open to suggestions as to which classes of equivariant neural networks the referee would suggest to us for future comparisons. A particular challenge in our considered settings of probabilistic prediction (the GP models being returning probability density functions, which is in turn  necessary for the log-score we are using in evaluation) is to come up with probabilistic approaches to equivariant neural networks, which is not something we found off-the-shelf and calls in our opinion for work that goes substantially beyond our present contributions.  We are by the way very thankful for the referee to point that “the theoretical characterization is rigorous and a strong point of the paper”.
>
> Concerning the choice of the fundamental region (and section/projection), while we provided an example at the end of section 5.1 highlighting the performance loss in case of a poorly chosen A, we will stress this further in the discussion and add new illustrations in appendix of how things can go wrong in particular cases with disconnected A’s (See the three figures on the topic added to the anonymous GitHub). Besides this, as developed in the response to referee U1Nr, we are not claiming the fundamental region approach to be uniformly better than others, but found it to be practically applicable in cases where integrations appeared to be prohibitively cumbersome.  As exposed in Appendix B2, we found it promising indeed to hybridize fundamental region (on an infinite group) and double sum (on a small group). We consider putting this more to the fore as this subtle combination may be extended to further contexts. Coming finally to remarks pertaining to sensitivity and robustness, we spent considerable energy working on hyperparameter tuning and arrived at reducing variability in implementation results. The choice of starting hyperparameter values plays an influential role, and to illustrate this and address the referee’s comment we conducted new experiments (available figure on anonymized Github) that will be reflected in Appendix.  We truly hope that in light of our efforts, the referee will consider increasing their score. We are thankful for the opportunities given to us to clarify things and improve our paper with a broader perspective.

---

> > ### Comment · Reviewer_HxJv · 2025-04-02
> >
> > Thank you for for clarifying. My concerns were solved and I will raise my score.

---

### Decision · Program_Chairs · 2025-05-01

**Decision:**

Accept (poster)

**Comment:**

Four knowledgeable reviewers recommend Accept and I accept. The authors should incorporate several clarification comments mentioned by the different reviewers.